



# Combining empirical and mechanistic understanding of spruce bark beetle outbreak dynamics in the LPJ-GUESS (v4.1, r13130) vegetation model

Fredrik Lagergren[1], Anna Maria Jönsson[1], Mats Lindeskog[1], and Thomas A. M. Pugh[1,2,3]

[1]Department of Physical Geography and Ecosystem Science, Lund University, Lund, Sweden.
[2]School of Geography, Earth and Environmental Sciences, University of Birmingham, Birmingham, UK.
[3]Birmingham Institute of Forest Research, University of Birmingham, Birmingham, UK.

*Correspondence to*: Fredrik Lagergren (Fredrik.lagergren@nateko.lu.se)

**Abstract.** For evaluating the forests' performance in a future with changing climate for different management alternatives, dynamic vegetation models are important tools. One of the functions in such models that has a big influence on the results is tree mortality. Bark beetles are important for the pattern of mortality in forest, especially for needle leaved forest in the temperate and boreal zones. The European spruce bark beetle (SBB, *Ips typographus*) has in the most recent years replaced wind as the most important disturbance agent in European forests. Historically, SBB damage is typically triggered by wind storms as they create breeding material with no defences to overcome for the beetles. Drought can contribute to increased damage and prolonged outbreaks by lowering the defence of the trees, but has been the main driver of some of the European forest damage in the last decade. In this study we implemented a SBB damage module in a dynamic vegetation model (LPJ-GUESS) that includes representation of wind damage and forest management. The module was calibrated against observations of storm and SBB damage in Sweden, Switzerland, Austria and France. An index of the SBB population size that changed over time driven by phenology, drought, storm felled spruce trees and density of the beetle population, was used to scale modelled damage. The model was able to catch the start and duration of outbreaks triggered by storm damage reasonably well but there was a large variability that partly can be related to salvage logging of storm felled forest and sanitary cutting of infested trees. The model showed increased damage in most recent years with warm and dry conditions, although below the level reported, which may suggest that the drought stress response of spruce in LPJ-GUESS is underestimated. The new model forms a basis to explore vulnerability of European forests to spruce bark beetle infestations.

## 1. Introduction

Intensified forest management, reforestation and fertilization effects from increased $CO_2$ concentration and nitrogen deposition have promoted the growth and stock of European forests over recent decades (Ciais et al., 2008; Scheel et al., 2022). However, over the most recent years this trend no longer exists, due to increased harvest and disturbances (Palahi et al., 2021; Patacca et al., 2022; Wernick et al., 2021). The disturbances and associated tree mortality can be related to weather and climate, directly or in combination with a biological agent, and the forest's structure and species composition is important as most epidemic species are selective in host preference (Balla et al., 2021). The prevailing paradigm of European forestry has increased the vulnerability of forests to disturbances by promoting monocultures (Forzieri et al., 2021). Forest policy in Europe is determined both at the national and European Union levels. Tools that can provide projections of how disturbances are likely to change





under different climate conditions and management actions at these national-to-continental scales are important to be able to develop appropriate forest policy.

Bark beetles are a group of insect species that in most cases colonize dead or stressed trees. A few species are, however, also able to kill healthy trees, potentially leading to outbreaks that can cause severe mortality, especially in temperate and boreal needle-leaved forests (Kautz et al., 2017; Lindgren and Raffa, 2013). In Europe, the most damaging bark beetle is the spruce bark beetle (*Ips typographus*, hereafter referred to as SBB), causing tree death corresponding to tens of million m³ wood per year (Patacca et al., 2022). The most recent years have seen accelerating losses of trees from SBB outbreaks, mainly related to droughts reducing the capacity of trees to defend themselves and climate warming accelerating the lifecycle of the beetles (Hlásny et al., 2021). Historically, however, outbreaks have mainly been triggered by storm damage, which creates breeding material in which the beetles are not required to overcome the tree defence (Schroeder, 2001). Norway spruce (*Picea abies*), the host tree, is favoured by forestry and is often managed with thinning and clear-cuts that creates dense even-aged stands, that over time becomes increasingly vulnerable to both storm damage and SBB attacks. The interaction of these multiple drivers makes SBB damage challenging to model at any scale. Whilst landscape- and national-level models for SBB exist (De Bruijn et al., 2014; Jönsson et al., 2012; Seidl et al., 2014), the capability to explicitly model forest damage from SBB is not currently available in a European-scale forest model.

The life cycle and temperature dependent phenology of SBB are well studied (Wermelinger and Seifert, 1998). However, to be able to evaluate model simulations on spruce bark beetle population dynamics and the shift between endemic and epidemic conditions, not only phenological understanding is needed, but also detailed knowledge about forest conditions. This includes stand composition and structure, timing and magnitude of storm damage, and subsequent management actions such as sanitary and salvage cutting (Jönsson et al., 2012). So far, empirical approaches have been applied in SBB modelling at continental (Marini et al., 2017) and regional scale (Soukhovolsky et al., 2022). Most empirical models include response functions of storm felling, temperature and precipitation deficit, but require prior knowledge of the SBB population, as the output from such models is a relative change in SBB population or damage, limiting the simulation of full outbreak cycles. In detailed mechanistic models, the timing of salvage and sanitary cutting is an important component (Jönsson et al., 2012), which makes data availability a limitation to large-scale model calibration. A model without mechanistic responses of climate and forest structure may, however, not be able to project future conditions and the effect of adaption in proactive and reactive forest management strategies. Conversely, the SBB lifecycle has several temperature dependent stages for reproduction and survival, which makes representation by a continuous response function, as typically employed by empirical models, difficult. Yet, when working over large regions with gridded climate datasets, the weather variability within a grid cell related to altitude, aspect and forest edge effects smooth out some of the non-linearity. This means that an empirical approach can be efficient in applications over large regions, also when considering the highest resolution of gridded climate data available today (typically 3 km, e.g. Lind et al., 2020). An approach that blends the insight of mechanistic modelling with the ability of large-scale empirical assessments to capture the net effects of unrepresented smaller-scale processes is needed.

Here, we seek to develop and evaluate a model for forest damage from SBB outbreaks within the LPJ-GUESS dynamic vegetation model (Smith et al., 2001; Smith et al., 2014). Because the model will be applied at relatively large scales with a focus on carbon cycling, forest productivity and resilience, the aim is to accurately characterise



the behaviour of the system, rather than predict specific events. For instance, it is not necessary to capture the particular year that specific outbreaks occur, but rather the typical magnitude and frequency of large outbreak events, as well as the typical background rates. It is, however, important to be able to simulate the impacts of management interventions such as salvage and sanitary cutting, and systematic changes in forest structure.

Furthermore, because SBB is only one of many bark beetles that have large impacts on forest dynamics across northern forests (Kautz et al., 2017), there is also a need to develop a model structure that is flexible to simulate a category of pest with multiple bark beetle species with different characteristics across a wide variety of forest across the temperate and boreal forest biomes. The aim of this study was therefore to develop a model with the following characteristics:

1. To be able to catch outbreak dynamics, triggered by storm damage, drought stress and temperature-driven changes in beetle phenology.
2. To utilise simple empirical relationships where available, but also make use of suitable mechanistic knowledge that is relevant at the scale of interest.
3. To capture the general outbreak dynamics without detailed accounting of SBB population.
4. To generate fractions of killed individuals in vegetation in tree size/age cohorts for feedback to modelled vegetation dynamics
5. To provide a generalised concept to use for different types of bark beetles by changing the underlying functions related to weather and insect – host tree interactions.

**2. Material and Methods**

**2.1. General description of LPJ-GUESS**

LPJ-GUESS is a dynamic vegetation model adapted to both global and European simulation domains, which simulates the development of vegetation cohorts belonging to different plant functional types (PFT) in replicate patches (1000 $m^2$) (Smith et al., 2001; Smith et al., 2014). The cohorts compete for water, nutrients and light within the patches, driven by climate, nitrogen deposition and atmospheric $CO_2$ concentration. The PFTs differ in

parameters related to physiological response functions, allometry and bioclimatic restrictions. Replicate patches are subject to the same climatic and edaphic conditions, but differ due to stochastic elements in the processes of cohort establishment and individual tree mortality and disturbance. A range of different forest management classes can be simulated in each model grid cell, for instance clear-cut or continuous harvesting following a particular regime or unmanaged vegetation (Lindeskog et al., 2021). Within these classes, there may be patches of different

ages since the last patch-destroying (i.e. stand-replacing) natural disturbance or clear-cut harvest event. Processes in the model includes e.g. light absorption, photosynthesis, auto- and heterotrophic respiration, allocation and different types of mortality. Patch destroying disturbances apart from fire have typically been simulated as random events with a fixed average return time.

In the present study LPJ-GUESS version 4.1, subversion revision 13130 was used. In addition to the standard

trunk version of the model, this revision also included the storm damage module from Lagergren et al. (2012). In this module, the simulated storm-damaged fraction of a cohorts is the product of a cohort's sensitivity index (SI), the triggering wind load (WL) and a calibration factor (CF).

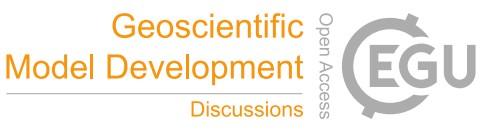
$$DF_{cohort} = SI_{cohort} \times WL \times CF \tag{1}$$

The SI is based on tree shape (height to diameter ratio), neighbouring cohorts' and stands' height and time since thinning.

**2.2. Setup of simulations representing the current state of spruce forest**

The global set of PFTs were used in the simulations, as the aim was to produce a biotic model that could be applied routinely in the standard version of LPJ-GUESS. This set has seven boreal and temperate tree PFTs, of which spruce is represented by the shade tolerant boreal needle leaved evergreen tree type (BNE). The spruce forest for each region/country was represented in the simulations by nine age classes of clear-cut forestry with a 90-year rotation period established at 10-year intervals after year 1859, continuous-cover forestry (CCF) with short cutting interval (12 years, 15% of biomass removed at each occasion) and long (25 years, 30%) and unmanaged forest. For the clear-cut rotations thinning was done at age 9, 27 and 45 years with strength of 10%, 30% and 25% of biomass removed. Planting or establishment in the managed forest types were set to BNE only. The results were weighted by weights based on the age-class distribution from Poulter et al. (2019) for year 2010. The data had a regional resolution for France and a national for Austria and Switzerland. For Sweden national inventory data for 2008-2012 were used instead (Skogsdata 2013, https://www.slu.se/en/Collaborative-Centres-and-Projects/the-swedish-national-forest-inventory/foreststatistics/skogsdata/). The age classes 91-110 and 111-140 years were represented by short and long CCF respectively, and potential natural vegetation (PNV) represented forest older than 140 years. The weights, related to the age in 2010, were used as input to the landcover functionality in LPJ-GUESS (Lindeskog et al., 2021) and were held constant over the simulations. Each age class, CCF type and natural forest was run for five replicate patches in each grid cell. Random patch-destroying disturbances were initially set to a return time of 500 years, but after the introduction of management in a patch, these were turned off to avoid resetting the age structure. In Europe, BNE also represents silver fir (*Abies alba*), which is not attacked by SBB, this could potentially lead to overestimation of SBB damage in stands simulated as unmanaged, as BNE in those stands would represent both species. In Switzerland that has the highest fraction of PNV (25%), silver fir only makes up 6.6% of the total spruce and fir standing volume. In the French regions there are 0-12% PNV, but here silver fir makes up 49% of total fir and spruce. Austria has only 7% PNV and in Sweden there are no silver fir. Since PNV multiplied by fir fraction was low (<6%) in all regions/countries, we ignored this problem. Furthermore, as Picea abies is the primary species constituting the BNE PFT within Europe, the parameterisation here can also be considered applicable in the European version of the model (Pugh et al., Manuscript).

**2.3. Implementation of a bark beetle damage module in LPJ-GUESS**

European storm and bark beetle damage statistics were compiled by Marini et al. (2017), and they used the dataset to derive empirical models describing the increase rate ($R$) of forest volume loss due to bark beetles ($D_{SBB}$) to one year ($t$) from the previous year ($t$-1):

$$R = log_e(D_{SBB\,t}/D_{SBB\,t-1}) \tag{2}$$

The top-rated model was:





$$R = -0.099 + 0.223T_t + 0.265D_{\text{storm } t-1} - 0.351D_{\text{SBB } t-1} - 0.151W_{t-1} - 0.052W_t - 0.233T_tD_{\text{storm } t-1} +$$
$$0.153T_tW_{t-1} + 0.233D_{\text{storm } t-1}W_t \; , \qquad (3)$$

depending on the thermal sum between 1 May and 30 July ($T$) with a threshold of + 5° C, storm felled volume ($D_{\text{storm}}$), $D_{\text{SBB}}$, and cumulative rainfall between 1 March and 31 July ($W$). All variables were standardized to mean 0 and standard deviation 1, $D_{\text{storm}}$ was log-transformed before standardization.

Using this concept, we implemented an additive model for bark beetle damage in LPJ-GUESS, taking advantage of the existing LPJ-GUESS formulations for water stress and mortality of trees ($L$, stem litter from spruces larger

than a limit ($d_{\text{lim}}$) killed by other agents than bark beetles last year), and the bark-beetle phenology of Jönsson et al. (2007).

$$R = f\left(P_{\text{gridcell } t-1}\right) + f(P_{\text{patch } t-1}/L) + f(\text{water stress}) + f(\text{phenology}) \qquad (4)$$

$$P_{\text{patch}} = e^R P_{\text{patch } t-1} \qquad (5)$$

$$M = P_{\text{patch}}k_0 = e^R P_{\text{patch } t-1}k_0 \qquad (6)$$

In which an index of the population size at the start of the year ($P_{t-1}$), calculated both at patch and grid-cell level, determines the mass of bark-beetle killed trees ($M$) together with $R$ and a calibration factor ($k_0$). Of the components of $R$ (Fig. 1), $f(P_{\text{patch}} / L)$ represents the negative feedback from a denser population relative to the amount of substrate with no defence ($L$) for a group of trees (patch), $f(P_{\text{gridcell}})$ represent the negative feedback when a high population in the landscape leads to lower patch level $R$ because of swarming induced competition, $f(\text{water stress})$

has a positive impact on $R$ as the defence in healthy trees is reduced with water limitation, and $f(\text{phenology})$ has a positive impact resulting from faster phenological development of the SBB.

The fraction of the different age and management classes, from the landcover functionality in LPJ-GUESS, were used to calculate $P_{\text{gridcell}}$ weighted over the classes. The possible range of the different parts of the model were given weights of the same magnitude as in the (Marini et al., 2017) model. The range of the components were

adjusted to give a total maximum range of $R$ (Fig. 1a) comparable to $R$ calculated (Eq. 2) from the damage statistics data (see section 2.4 below). The maximum $R$ can also be translated to the population increase rate with two successful generations in a year with 21 female offspring per mother ($e^{6.1} = 21.1^2$). To enable that an outbreak can also be sustained at the highest population levels, the lowest possible total negative feedback from population size is just below the highest possible positive feedback from water stress and phenology.




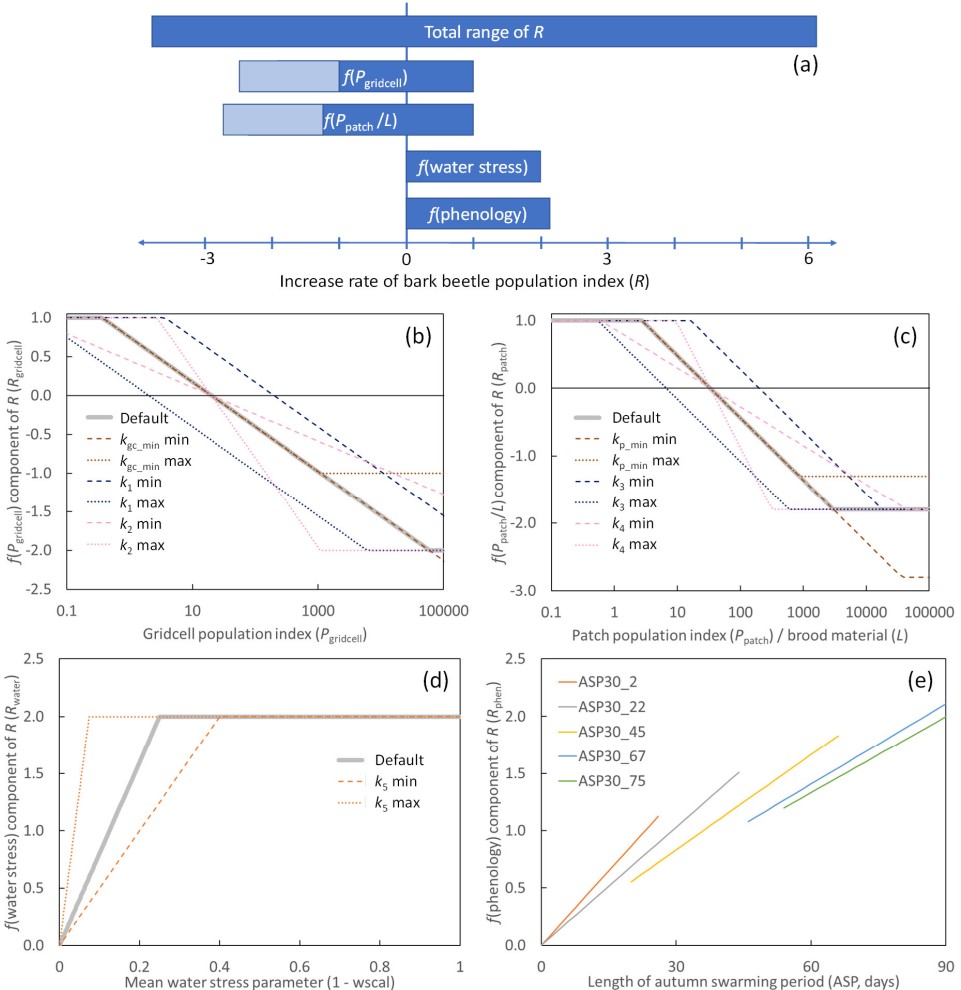

**Figure 1: The different components of the increase rate of the bark beetle population index (R). (a) Ranges of the components and the total. For $f(P_{gridcell})$ and $f(P_{patch}/L)$, the light shaded areas show the range with different parameters for the minimum of $f(P_{gridcell})$ and $f(P_{patch}/L)$. To have the possible total negative feedback from the population index constant, the minimums of the ranges were dependent on each other, such that the sum always equalled -3.8. (b-e) Shape of the functions for the components of the increase rate of the bark beetle population index (R). The default parameter setting is shown by thick grey lines (b-d). The functions are also shown in colour for the min and max value of parameters included in the calibration and sensitivity analysis, using the default setting for the other parameters. For $f(phenology)$, (e), no parameters were tested but the response depends on the grid-cell's 30-year running mean of the length of the autumn swarming period (ASP, ASP30) and function are shown for ASP30 from 2 to 75 days.**

The response function for the $P_{gridcell}$ component of $R$ (Fig. 1b) was:

$$R_{gridcell} = \mathrm{MIN}(k_{gc\_max}, \mathrm{MAX}(k_{gc\_min}, -\log_e(P_{gridcell} \times k_1) \times k_2)), \qquad (7)$$

where $k_{gc\_max}$ and $k_{gc\_min}$ determine the range, $k_1$ the intercept and $k_2$ the slope of the expression. The same type of function was used for the combined response of $P_{patch}$ and $L$ (Fig. 1c):



$$R_{\text{patch}} = \text{MIN}(k_{\text{p\_max}}, \text{MAX}(k_{\text{p\_min}}, -\log_e(P_{\text{patch}}/L \times k_3) \times k_4)), \tag{8}$$

where the min and max are set by $k_{\text{p\_max}}$ and $k_{\text{p\_min}}$ respectively, and $k_3$ and $k_4$ set the intercept and slope. To avoid division by zero, a fixed background level ($k_{\text{base\_bm}}$, set to 0.0001 kg C m$^{-2}$) was added to the available brood material ($L$):

$$L = k_{\text{base\_bm}} + L_{\text{mort}}, \tag{9}$$

where $L_{\text{mort}}$ is C mass of stem mortality of spruce trees above a diameter threshold ($d_{\text{lim}}$) for previous year caused by other reasons than bark beetles (including storm), $d_{\text{lim}}$ was set to 15 cm (Jönsson et al., 2012). The ratio between water supply to the canopy and canopy water demand (wscal), as calculated by LPJ-GUESS, that goes from zero at complete shutdown of photosynthesis and transpiration to one at no stress, was used to assess the dependency

of drought (1 - wscal, Fig. 1d):

$$R_{\text{water}} = \text{MIN}(k_{\text{drought\_max}}, (1 - \text{wscal}_{\text{mean}}) \times k_5), \tag{10}$$

where $k_5$ is the slope, determining the point of full effect on $R$. The mean wscal was calculated over the month May-July for both previous year and current year, using data for BNE. Based on the (Marini et al., 2017) model, the data from the previous year were given a three times higher weight in the default setting, but in the calibration

and sensitivity analysis (see below) the previous year weight ($k_{\text{pyw}}$) was varied between ¼ to 4. For a more mechanistic approach of taking phenology into account, the dependency of $T$ in the (Marini et al., 2017) model was further developed to express the inter-annual variation in length of the autumn swarming period (ASP) in comparison with the grid-cell specific 30-year average (Jönsson et al., 2011) as:

$$R_{\text{phen}} = \text{ASP} \frac{k_6}{\text{ASP30} + k_7}, \tag{11}$$

where the slope depends on the grid-cell-level 30-year running mean of ASP (ASP30) and two parameters $k_6$ (slope) and $k_7$ (dampening). Calculated ASP was capped at 90 days (ASP$_{\text{max}}$). Normalizing with ASP30 instead of a direct relationship ($R_{\text{phen}} = \text{ASP} \times 2 / \text{ASP}_{\text{max}}$) gives a more responsive function at lower ASP30 (Fig. 1e).

In the managed forest of Europe, countermeasures against outbreaks are often performed. We included functionality of salvage cutting of storm-felled trees and sanitary cutting of infected trees (SSC). The salvage

cutting part was done by reducing $L_{\text{mort}}$ by 90% for larger storm events (>5 m$^3$ wood at patch level). The total grid-cell maximum salvage cutting capacity (salvmax) was set to 50% of the 10-year average harvest rate, as it has to be done before the new generation of bark beetles emerge (which occurs approximately 6 months into the year) in order to have an effect. If the 90% of the storm damage (damage_available) was > salvmax, $L_{\text{mort}}$ was reduced by salvmax/damage_available instead of 90%. Sanitary cutting was applied by reducing $P_{\text{patch}}$ by 25% if

the fraction of available spruce volume that would be killed was >1%. This reduction was done before Eq. (6) was applied.



### 2.4. Data of storm and bark beetle damage to forest in Europe

Data of damaged volume of spruce forest were combined with statistics of standing spruce volume to assess the fractions in a country/region damaged by storm ($DF_{storm}$) or killed by SBB ($DF_{SBB}$), which were calculated and
used for further analysis.

European storm and bark beetle damage statistics were compiled by Marini et al. (2017), for some countries separated into administrative or topographical units. From that data set we used damage statistics ($m^3 \ yr^{-1}$) from South Sweden (data separated into 10 counties), North-East France (five former administrative units), Switzerland (lowland and mountains) and Austria (whole country), to cover regions with large interannual variability in ASP.
As storms in Europe mainly occur in autumn and winter, and as the vegetation and bark beetle effect will be the same for a storm event in October or in February the next year, the storm damage statistics for a specific year were compiled for a storm season of July specified year until June the following year.

Data of spruce standing volume were available by year (1961-2010) for Sweden (https://www.slu.se/en/Collaborative-Centres-and-Projects/the-swedish-national-forest-inventory/long-time-
series/time-series-from-1953/), for year 2008 for France (https://inventaire-forestier.ign.fr/spip.php?rubrique250), for year 1985, 1995 and 2006 for Switzerland (https://www.lfi.ch/publikationen/publikation-en.php) and for 2008 in Austria (https://www.bfw.gv.at/en/departments-en/forest-inventory/). For the other countries than Sweden with data only for one or three years, the data were interpolated between the inventory years and kept constant before the first
and after the last year.

In recent years Europe has faced several SBB outbreaks that have been driven by warm and dry conditions rather than triggered by storm events (Nardi et al., 2023; Trubin et al., 2022). To test the impact of this shift in driving factor for the parameterization, national level storm and bark beetle damage statistics from the Standardized Disturbance Index (SDI) dataset (Patacca et al., 2022) 2011-2019 for Switzerland and Austria were used. For
Switzerland the damage was split between Lowland and Mountain assuming that the proportion of the country totals for Lowland and Mountain parts were the same as in the 1990-2010 data described above.

**Table 1: Summary of available storm and SBB damage data from Marini et al. (2017), data used in the primary calibration and additional data from SDI (Patacca et al., 2022) used in the validation as well as the number of simulated**
**climate grid-cells in the different parts.**

| Country or part of country | Regions | Gridcells per region | Primary calibration years (Marini et al., 2017) | Additional calibration years (Patacca et al., 2022) |
|---|---|---|---|---|
| South Sweden | 10 | 3-9 | 1990-2010 | |
| Switzerland | 2 | 8-9 | 1990-2010 | 2011-2019 |
| Austria | 1 | 39 | 1990-2010 | 2011-2019 |
| East France | 5 | 4-17 | 2000-2010 | |



Wind load (WL) calculated (Eq. (6) in Lagergren et al., 2012) from CRU wind data (see below), which is normally used to force to the wind damage module, was poorly related to the observed timeseries of wind damage ($DF_{storm}$, fraction of spruce forest damaged). Therefore, in order to focus on bark beetle outbreak dynamics without introducing additional uncertainties associated with wind damage modelling, we adjusted WL such that it followed $DF_{storm}$ (denoted $WL_{stat}$). This approach means that we take advantage of the wind damage module to distribute the damage at cohort level (Eq. 1) but, as general functions were used to go from $DF_{storm}$ to $WL_{stat}$, the exact $DF_{storm}$ time series will not be reproduced by the model. In a first step, a factor of 2 was found to approximately generate the same average level of $WL_{stat}$ calculated from $DF_{storm}$ as WL calculated from wind data for years 1990-2010.

$$WL_{stat} = DF_{storm} \times 2 \tag{12}$$

After evaluating the quotient between preliminary LPJ-GUESS simulation results and inventoried $DF_{storm}$ at regional level for all available years (Table 1), we concluded to use a separate function for northern Europe (Sweden) depending also on latitude (LAT):

$$WL_{stat} = DF_{storm} \times 2/f(LAT) \tag{13}$$

$$f(LAT) = -0.00412 \times LAT^2 + 0.425 \times LAT - 9.94 \, , \tag{14}$$

where Eq. (14) was fitted to $DF_{storm}$ quotients in S Sweden 1965-2010. The scaling with LAT, as a proxy for productivity (LAT explains 82% of the variation in county average site quality class in Southern Sweden and 92% for all counties in Sweden (Tab. 3.11a, Skogsdata 2023, https://www.slu.se/en/Collaborative-Centres-and-Projects/the-swedish-national-forest-inventory/foreststatistics/skogsdata/), is reasonable as a higher WL is needed to trigger a certain level of damage when the cohorts have a lower SI (as they have with lower productivity).

### 2.5. Climate data and area delimitation

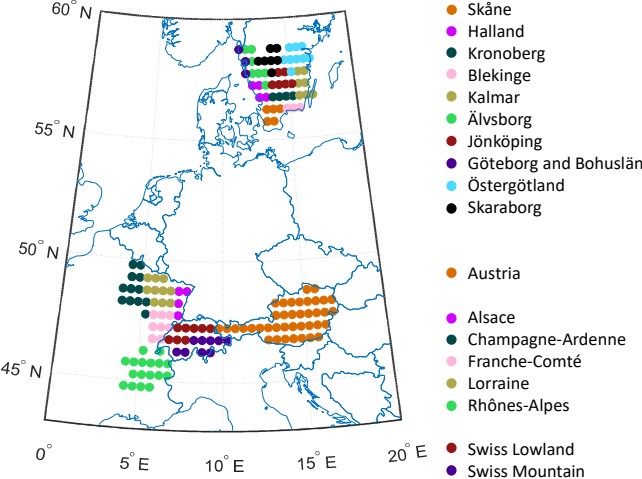

**Figure 2:** The climate gridcells simulated for the ten counties in South Sweden, the five counties in North-East France, the two larger regions in Switzerland and in Austria.



The simulations were driven by daily weather data 1901-2019 at 0.5° resolution from the CRU-JRA V2.1 dataset (https://catalogue.ceda.ac.uk/uuid/10d2c73e5a7d46f4ada08b0a26302ef7). For nitrogen deposition, monthly data from Lamarque et al. (2010) were used. The model was run with forest vegetation for all grid cells within each region/country (Fig. 2). For the years with no wind damage statistics, WL was calculated from CRU windspeed

from the cubed exceedance of the 99.5 percentile of daily wind speed accumulated over storm season as in Lagergren et al. (2012). Wind damage was applied from 1951 and forward in the simulations.

**2.6. Calibration and sensitivity test**

For calibrating and testing the parameter sensitivity of the model, data for 1990-2010 from the 10 southern counties in South Sweden (excluding the Gotland island, which has a low fraction of spruce forest and non-typical

soils), 5 counties in North-East France (only data for 1999-2010), Austria (whole country) and the lowland and mountain regions of Switzerland were used. For South Sweden, North-East France and Switzerland the modelling results and calibration data of $DF_{SBB}$ were averaged over the counties/regions for use in the calibration and sensitivity test. The parameters were first adjusted based on expert judgement, giving a default set of parameters (Table 2).


Table 2: Parameters in the bark beetle model. Min and max value shown for parameters used in the calibration and sensitivity analysis.

| Parameter | Part of model | Default | Min | Max |
|---|---|---|---|---|
| $k_{gc\_min}$ | $f$(Pgridcell) | -2.0 | -2.5 | -1.0 |
| $k_{gc\_max}$ | $f$(Pgridcell) | 1.0 | | |
| $k_1$ | $f$(Pgridcell) | 0.05 | 0.005 | 0.5 |
| $k_2$ | $f$(Pgridcell) | 0.25 | 0.15 | 0.5 |
| $k_{base\_bm}$ | $f$(Ppatch / L) | 0.0001 | | |
| $k_{p\_min}$ | $f$(Ppatch / L) | -1.8 | -2.8 | -1.3 |
| $k_{p\_max}$ | $f$(Ppatch / L) | 1.0 | | |
| $k_3$ | $f$(Ppatch / L) | 0.03 | 0.005 | 0.15 |
| $k_4$ | $f$(Ppatch / L) | 0.4 | 0.25 | 0.8 |
| $k_{pyw}$ | $f$(water stress) | 3 | 0.25 | 4 |
| $k_{drought\_max}$ | $f$(water stress) | 2.0 | | |
| $k_5$ | $f$(water stress) | 8 | 5 | 27 |
| $k_6$ | $f$(phenology) | 3.33 | | |
| $k_7$ | $f$(phenology) | 75 | | |

Eight of the 14 parameters were then selected for a calibration procedure and a sensitivity test. These selected

parameters were mainly related to the shape of the functions; other parameters were not included as we wanted to keep the range of the response functions at approximately the same magnitude as the weight of the original Marini et al. (2017) model (Eq. 3). To further reduce the number of calculations and to keep the total weight of the population size dependency constant, the sum of the minimum of the ranges $f(P_{gridcell})$ and $f(P_{patch} / L)$ was kept constant ($k_{gc\_min} + k_{p\_min} = -3.8$). This meant that the number of tested parameters could be reduced to 7 as $k_{p\_min}$





could be replaced with -3.8 - $k_{gc\_min}$. For each tested parameter 7 discrete numbers were used, evenly spread
between the ranges in Table 2. LPJ-GUESS was first run with the default parameter setting. Output at cohort level
of all variables used in the bark beetle module was produced for this simulation. These data were then applied in
a stand-alone version of the bark beetle module implemented in Matlab (R2018b) for all $7^7$ parameter
combinations. This approach misses the feedback from bark-beetle mortality to the vegetation state that is
simulated within LPJ-GUESS, but greatly reduces the calculation time. Each parameter combination was first run
with the default $k_0$, $k_0$ was then iteratively adjusted until the mean quotient between simulated and observed
maximum $DF_{SBB}$ for the calibration period over the four regions/countries equalled 1. The results were sorted by
$R^2$, root mean square error (RMSE) and absolute bias both at country level (arithmetic mean over the 1-10 regions,
Table 1) and by arithmetic mean over the four countries' mean values, and the top ranked parameter combination
was selected based on highest $R^2$, lowest RMSE and lowest absolute Bias. A combined statistic measure (CSM)
was also calculated for all $7^7$ models ($X$) by summing normalized $1 - R^2$, RMSE and absolute Bias as:

$$\text{CSM}_X = \frac{(1-R_X^2)-(1-R_{mean}^2)}{(1-R_{max}^2)-(1-R_{min}^2)} + \frac{\text{RMSE}_X-\text{RMSE}_{mean}}{\text{RMSE}_{max}-\text{RMSE}_{min}} + \frac{|Bias|_X-|Bias|_{mean}}{|Bias|_{max}-|Bias|_{min}},\tag{15}$$

where the normalization is based on deviation from mean relative to the range (max-min). The $\text{CSM}_X$ values were
sorted in ascendant order and we present parameters and statistics for the highest rated model as well as the range
of parameters for the 50 highest rated models.

LPJ-GUESS was run both with and without SSC, then the parameter testing and calibration procedure was
repeated both with and without inclusion of the 2011-2019 calibration data for Switzerland and Austria. As a final
test of the impact of SSC for the results, the stand-alone implementation was also run with inclusion of SSC for
the parameter set obtained without SSC and vice versa. The setting including SSC using only the 1990-2010 data
was considered as the main base run, results from the other runs are in most cases presented in the Supplement.

**2.7. Exploring the climate change signal**

To test the robustness of the approach to test the model for different parameter combinations with structure and
$L_{mort}$ prescribed from an LPJ-GUESS simulation with default parameters, LPJ-GUESS was finally run with the
optimized parameter set, with feedback of the damage associated with that setting to the simulated vegetation. A
simple test of climate sensitivity was also done for this setting by applying +2 °C to the climate data throughout
the simulation.

**3. Results**

**3.1. Model optimization**

The top ranked set of model parameters differed depending on the country or region assessed (Table 3), the
calibration period included and whether SSC was included or not (Table S1). Most of the parameters of the
common model with the main base-run optimization for all four regions (with SSC and not including calibration
data for Switzerland and Austria 2011-2019) had a large span within the 50 highest ranked models (Table 3) but
only one parameter value or a narrow span was dominant (Fig. S1). It should be noted that the calibration always





was based on data from all countries, also when the optimum model for the regions countries was selected, which

explains why there is a difference for S Sweden and NE France between Table 3 and Table S1a and between Table S1b and Table S1c.

**Table 3: Parameters in the main base run setting (with SSC and not including calibration data for Switzerland and Austria 2011-2019), for the top-ranked model in terms of combined statistics of bias, RMSE and $R^2$ for all four regions/countries together and for each region/country separately. For all four regions/countries together, also the parameter range for the 50 highest ranked combinations is shown (min_50, max_50). The $k_0$ values are the results of the calibration, LPJ-GUESS was run with a $k_0$ of 0.003. The numbers in parenthesis is the order number (from smallest to largest) of the seven values tested within the full parameter range (Table 2).**

|  | $k_1$ | $k_2$ | $k_{gc\_min}$ | $k_{pyw}$ | $k_3$ | $k_4$ | $k_5$ | $k_0 \times 1000$ |
|---|---|---|---|---|---|---|---|---|
| Default | 0.05 (4) | 0.25 (3) | -2 (3) | 3 (6) | 0.03 (4) | 0.4 (4) | 8 (3) | 2.89 |
| All four | 0.005 (1) | 0.5 (7) | -1.75 (4) | 0.25 (1) | 0.15 (7) | 0.8 (7) | 5 (1) | 5.12 |
| min_50 | 0.005 (1) | 0.15 (1) | -1.75 (4) | 0.25 (1) | 0.005 (1) | 0.5 (5) | 5 (1) | |
| max_50 | 0.5 (7) | 0.5 (7) | -1 (7) | 4 (7) | 0.15 (7) | 0.8 (7) | 20 (6) | |
| | | | | | | | | |
| S Sweden | 0.5 (7) | 0.15 (1) | -2 (3) | 0.5 (3) | 0.009 (2) | 0.8 (7) | 27 (7) | 0.0989 |
| Switzerland | 0.005 (1) | 0.15 (1) | -2.25 (2) | 1 (4) | 0.005 (1) | 0.25 (1) | 6 (2) | 0.192 |
| Austria | 0.12 (5) | 0.15 (1) | -2.5 (1) | 1 (4) | 0.15 (7) | 0.5 (5) | 27 (7) | 0.00272 |
| NE France | 0.005 (1) | 0.35 (4) | -2.5 (1) | 4 (7) | 0.15 (7) | 0.4 (4) | 20 (6) | 0.0428 |

## 3.2. Model performance

The optimization procedure resulted in reduced bias and RMSE and increased $R^2$ compared to the default setting (Table 4, Fig. 3, Table S2, Fig. S2-4).

**Table 4: Statistics for different parameter settings; default (Table 2), the top-ranked model in terms of combined statistics of bias, RMSE and $R^2$ for all four regions/countries together and for each region/country separately, in the main base run.**

|  | Default | | | Combined all four | | | Region/country | | |
|---|---|---|---|---|---|---|---|---|---|
|  | $R^2$ | RMSE | Bias | $R^2$ | RMSE | Bias | $R^2$ | RMSE | Bias |
| All four | 0.30 | 0.29% | 0.10% | 0.40 | 0.22% | 0.04% | | | |
| S Sweden | 0.54 | 0.28% | 0.21% | 0.76 | 0.12% | 0.07% | 0.72 | 0.08% | 0.01% |
| Switzerland | 0.34 | 0.31% | -0.02% | 0.43 | 0.38% | 0.07% | 0.45 | 0.34% | 0.01% |
| Austria | 0.03 | 0.20% | 0.10% | 0.19 | 0.18% | 0.08% | 0.26 | 0.15% | 0.07% |
| NE France | 0.29 | 0.59% | -0.30% | 0.21 | 0.64% | -0.28% | 0.44 | 0.60% | -0.08% |

The model captured the outbreak dynamics well for S Sweden. In Switzerland $R^2$ was rather low, mainly because the peak in damage after the 1999 storm in 2001 was not captured. In North-East France there was a large spread

in the observed outbreak after the 1999 storm, with Alsace and Lorraine having a large peak in 2001, while the other counties had a more slowly evolving progress. In Austria the outbreak starting in 2003 after the 2002 storm, with significant storm damage also in 2008 and 2008, lasted almost a decade though the response from the



functions of phenology and drought did not indicate that these factors supported the outbreak, resulting in low $R^2$.
Including years 2011-2019 data from Austria and Switzerland in the calibration resulted in similar outbreak

pattern (Fig. 3, Fig. S2).

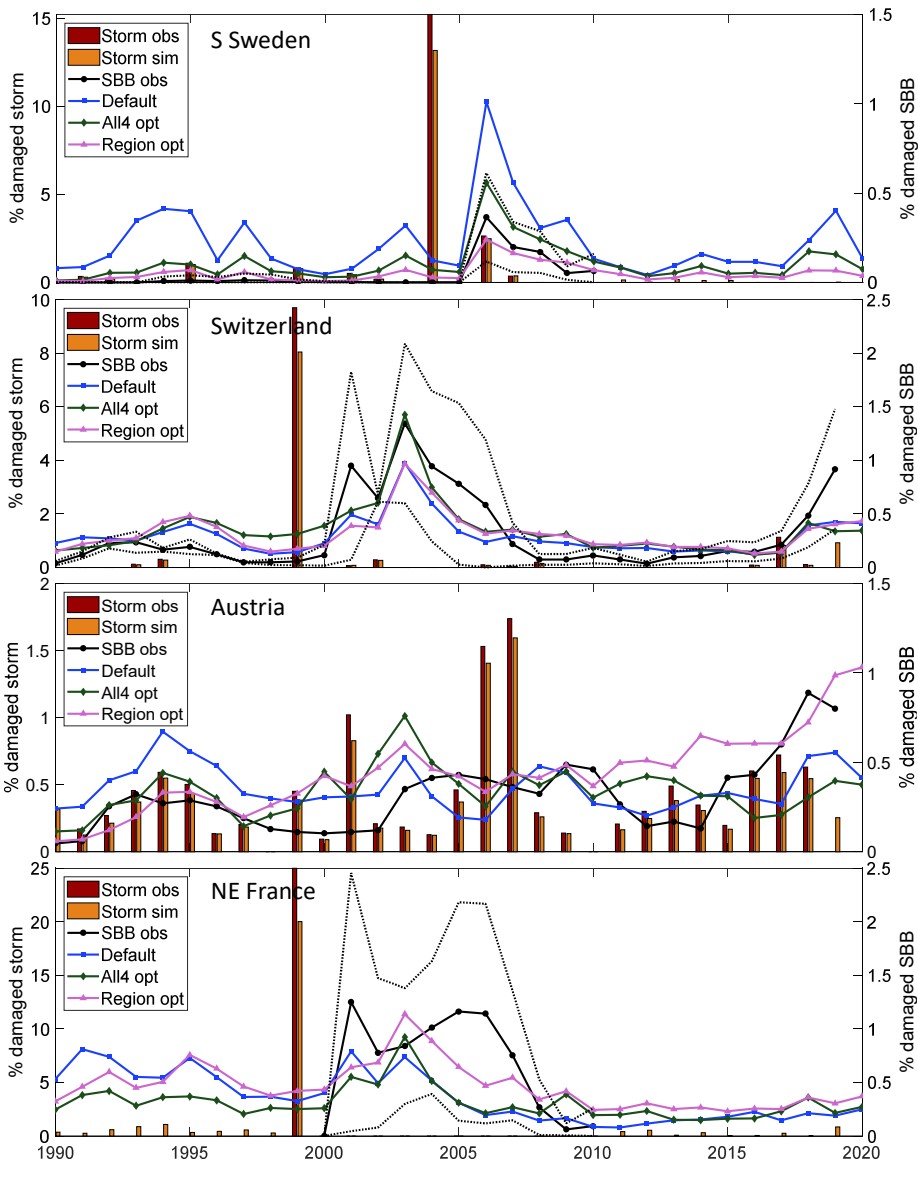

**Figure 3: Observed and modelled fraction of spruce forest damaged by storm (left y-axis) and SBB (right y-axis) in four regions/countries, with modelled SBB damage from different parameter settings (Table 3) in the main base run.**
**For Sweden (n = 10), Switzerland (n =2) and France (n = 5) +/- standard deviation between regions in observed SBB damage is shown with dotted lines.**



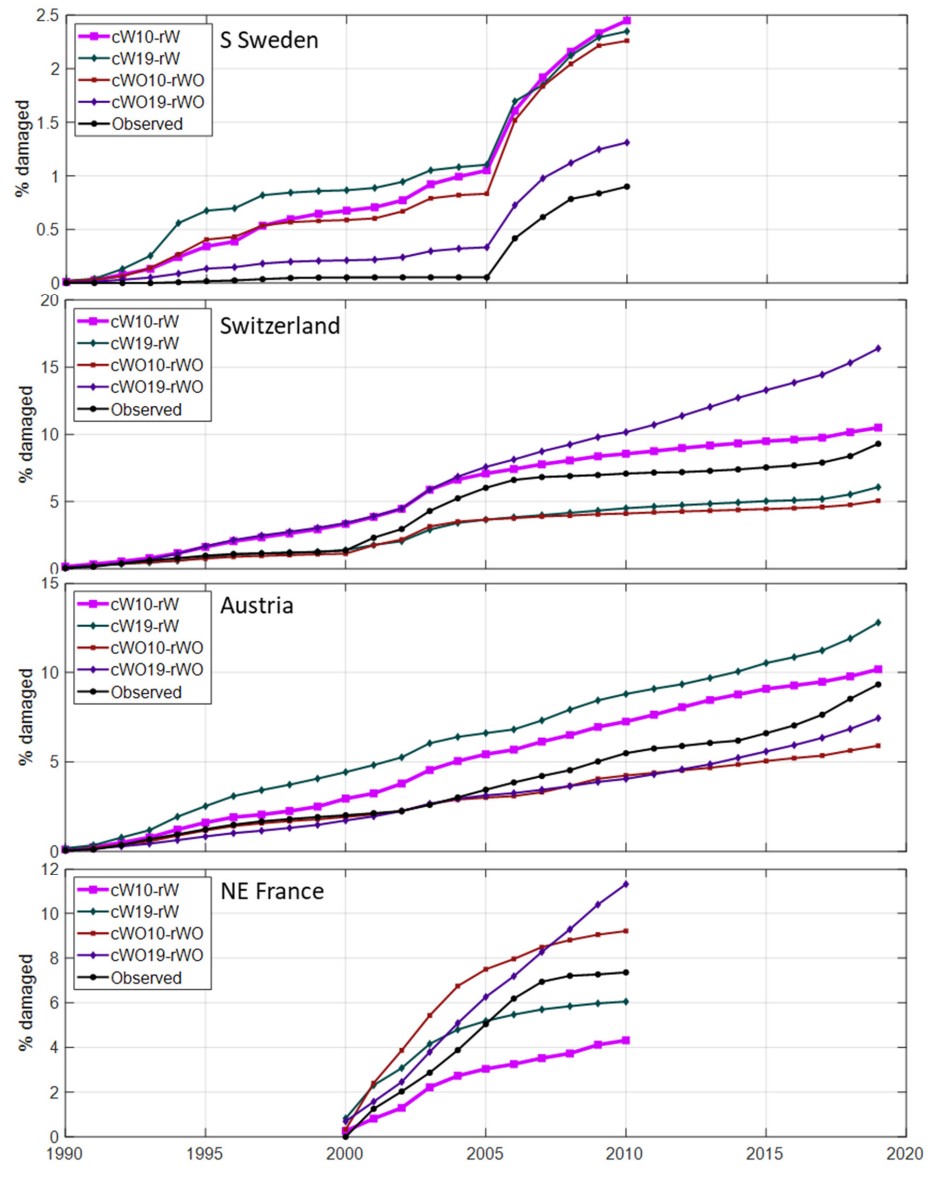

**Figure 4: Accumulated fraction of simulated and observed spruce forest damaged by SBB over the period with observations. Calibrations are shown using only data up until 2010 (10) and also including data for Switzerland and Austria 2011-2019 (19) as well as with (W) or without (WO) salvage and sanitary cutting. The main base run setting, cW10-rW, is showed by bold line.**

Accumulating the SBB damage over time show that level of damage during outbreak situations was generally well captured for all four region/countries (Fig. 4). The difference depending on setting for the calibration was quite large, and in the main base run the total damage for SE France was underestimated. During non-outbreak



situation in Sweden the accumulated damage is higher than observed while it agrees better in Austria and Switzerland but this may to a large extend depend on the way damage is reported, as discussed in Sect. 4.

### 3.3. Model sensitivity

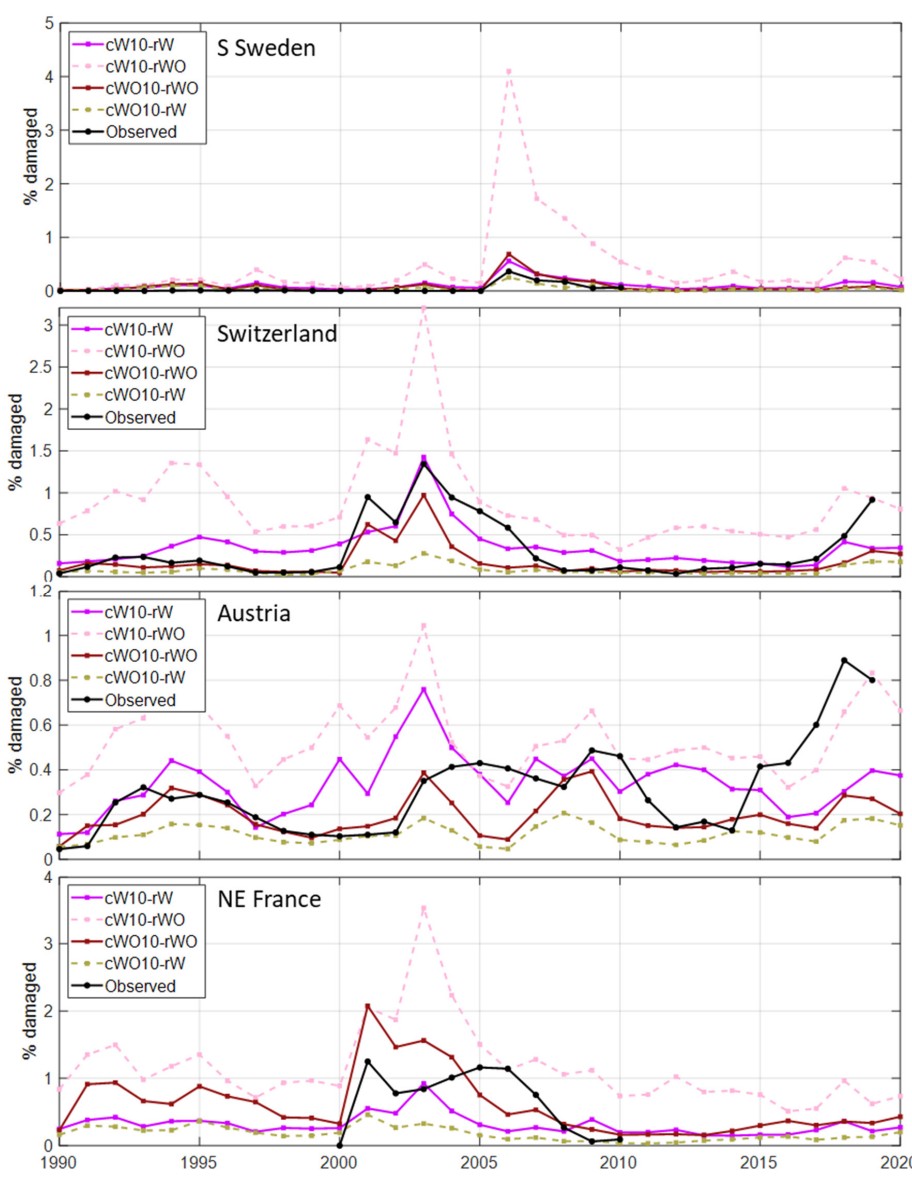

**Figure 5: Test of the sensitivity of including salvage cutting of storm felled trees and sanitary cutting of infected trees (SSC) for the fraction of SBB killed trees. The calibrations with (cW) and without (cWO) SSC using calibration data for 1990-2010 (10) were run both with (rW) and without (rWO) SSC. In Fig. S5, the calibrations also including 2011-2019 data for Austria and Switzerland are shown.**






Including the 2011-2019 data for Austria and Switzerland in the calibration resulted in higher accumulated SBB damage in Austria and NE France but lower level in Switzerland with SSC (Fig. 4). Without SSC it resulted in substantially higher accumulated damage in Switzerland and NE France as the outbreaks continued at a high level for more years (Fig. S3-4).

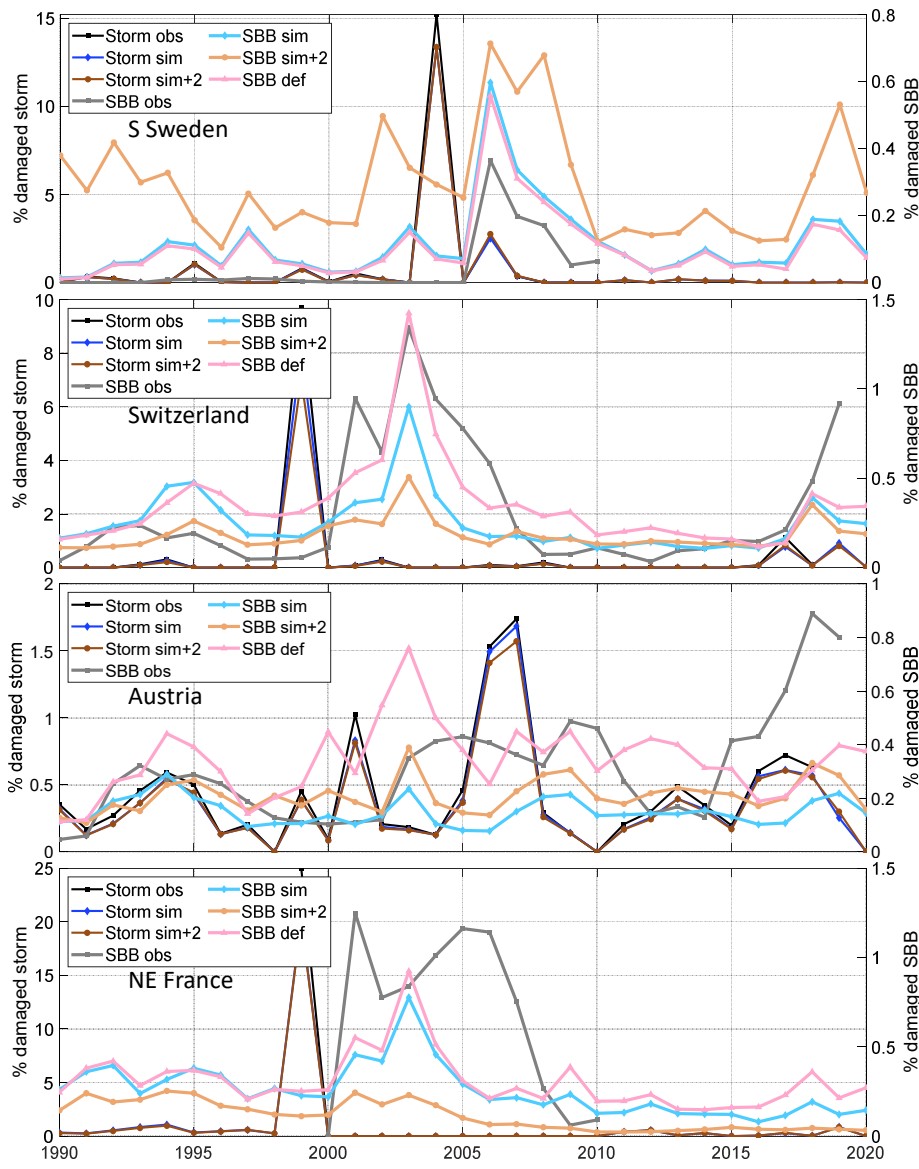


**Figure 6: Sensitivity test of modelled storm and spruce bark beetle damage for temperature and model environment. Model output from the calibrated stand-alone version with tree structure and storm damage input from the default LPJ-GUESS run (def, "All4 opt" in Fig. 3) and the calibrated model run with LPJ-GUESS with normal climate (SBB sim) and with +2 ° temperature (sim+2) compared to observations (obs).**





Applying the parameter set obtained with SSC without SSC resulted in 1.7 (Austria) to seven (Sweden) times higher simulated damage levels and the calibration without SSC run with SSC showed reductions of similar magnitude (Fig. 5, Fig. S5). It should though be noted that the negative feedback on forest vulnerability when trees are killed is not included in these stand-alone implementations, giving a higher background damage level in the run without SSC.

Up to this point all results are for the stand-alone Matlab implementation based on the vegetation from the default LPJ-GUESS run. Applying the calibrated parameter set of the main base run in LPJ-GUESS showed a very good match for S Sweden, a minor underestimation in NE France and underestimations by 35% and 70% in Switzerland and Austria, respectively (Fig. 6). An in-depth analysis for Austria showed that the difference was due to one grid cell where the missing negative feedback from spruce killing resulted in accumulated modelled damage that widely exceeded the availability of spruce trees in the stand-alone version. Removing this grid cell gave similarly low levels as for the LPJ-GUESS run with calibrated parameters, showing that the applied stand-alone optimization methodology resulted in an underestimation in this case.

The simple climate sensitivity test of increasing the temperature by 2 °C resulted in an expected increase in SBB damage in S Sweden and Austria (Fig. 6). In Switzerland and NE France, the warming resulted in significantly reduced biomass of the boreal BNE PFT and consequently a reduction in SBB damage.

**3.4 Predisposing, triggering and contributing factors**

The course of a of an insect outbreak can be boiled down to predisposing, triggering and contributing factors (Saxe, 1993). A summary of these factors by country are shown in Fig. 7, and will be the base for our discussion in Sect. 4.

**4. Discussion**

In this study, damage was assessed for the spruce fraction of the forest simulated for stands planted with the BNE PFT. The predisposition is, therefore, mainly determined by the amount of spruce stem volume with a diameter larger than 15 cm ($d_{\text{lim}}$), which depends on tree allometry, age-class distribution, growth rate and tree density. The diameter limit has no direct physiological relationship to SBB preference but is related to bark thickness, which is directly related to the possibility to breed (Schlyter and Anderbrant, 1993), with more bark beetle offspring per unit bark area in larger trees (Weslien and Regnander, 1990). In thinner trees there may be parts of the stem that has thick enough bark for successful breeding, but the beetles need a dense cover of galleries over a major part of the stem to overcome the defence. In larger trees the bark can be too thick at the base and the beetles attack higher up on the tree where a lower water potential in the stem (Tyree and Ewers, 1991) also contributes to the success of the attack. The tree density determines how quickly a tree reaches $d_{\text{lim}}$ and, in turn, depends on plant number, thinning and mortality. Age- and competition-related mortality also contributes to the background endemic SBB population, which is important for the level of damage when an outbreak is triggered. In this study, the size of age classes was determined for one point in time at country or county scale, which is a simplification. Likewise, the availability of spruce trees with $d > d_{\text{lim}}$ in a grid cell is based on age-class data for the year 2010. Currently available large-scale datasets do not intersect stand age and species composition, which induces a big uncertainty





when attempting to model outbreaks that are strongly dependent on the availability of trees of a particular species and size. A step forward in this respect could be to base the current stage on national forest inventories using plot level information on individual trees and use harvest and damage statistics for damage, thinning and clear-cut levels, which then in turn determine the tree composition and size distribution (Pugh et al., Manuscript).


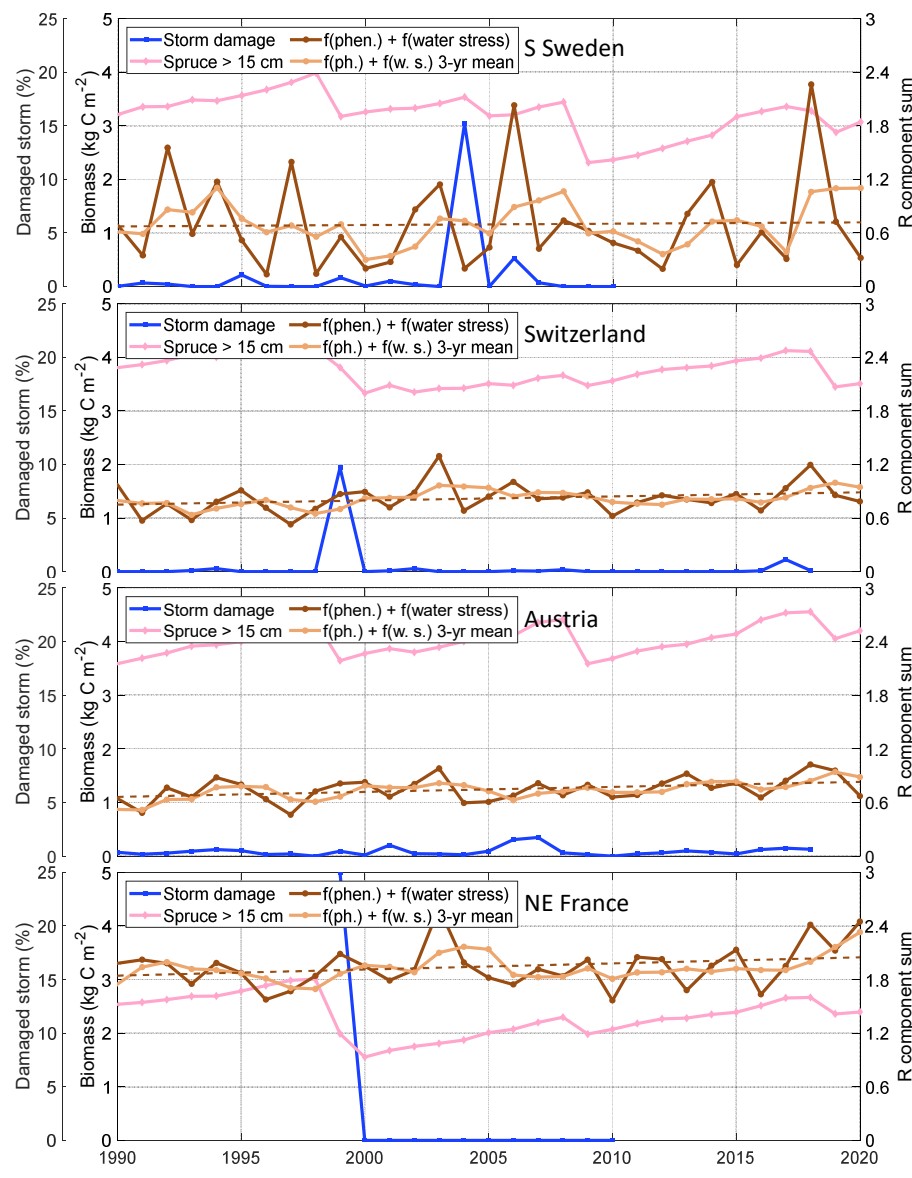

**Figure 7: Sensitivity for SBB outbreak divided in predisposing (biomass of spruce trees with a diameter larger than 15 cm), triggering (fraction of spruce volume damaged by storm) and contributing (the sum of the phenology and water stress related components for the increase rate of the bark beetle population index, *R* Eq. 4) factors. For the phenology and water stress *R* components sum the running 3-year mean and the trend over time (dashed line) are also shown.**






Bark beetles commonly develop epidemic levels during the growing season following a storm event ramping up the populations in the defenceless damaged trees. This then leads to killing of trees the second season after the storm. The timing of such outbreaks is quite well captured by the presented model. In managed forest, however,

the occurrence and timing of counter measures such as salvage logging of storm-felled trees, sanitary cutting of infected trees and insect traps have a very big impact on the outcome (Jönsson et al., 2012), factors that vary greatly in time and by region over Europe, as well as at finer scale (Wichmann and Ravn, 2001). By timely salvage logging, a large fraction of the beetles can be removed from the forest, substantially dampening the damage done. European-wide models calibrated over a longer time span (e.g. Marini et al., 2017) should not therefore be

expected to have a high explanation of particular events in comparison to more local model applications (e.g. Soukhovolsky et al., 2022). However, they should be able to capture the general characteristics of these events well. One possible route forward to improved evaluation of large-scale models is assembling local-scale datasets that combine high-resolution information on species composition and forest structure, with detailed information on bark beetle detection and sanitary/salvage logging. With on-going developments in remote sensing of forest

structure and composition combined with datasets on forest harvesting, this may be increasingly feasible in the near future (e.g. Jamali et al., 2024).

After a major storm, the remaining trees may have disturbed root systems making them more vulnerable to drought which may enhance SBB outbreaks, but also making them unstable – predisposing them to further storm damage. As SBB prefers defenceless trees even in an outbreak situation, such wind damage can serve as traps for the

beetles if the wood is salvaged in time. This situation occurred in Southern Sweden when the 2006-2007 storm dampened the outbreak following the major 2004-2005 storm. This relatively subtle mechanism is not captured in our model, which overestimated SBB damage in this outbreak. The outcome after a trigger depends on the initial level of the SBB population. Many SBB models require an initial population or damage level, which means that they work in relative terms (Marini et al., 2017; Soukhovolsky et al., 2022), but this is not feasible in a

dynamic forest model operating over large spatial scales.

The way damage is inventoried and reported may also be different between countries and over time. For example, the neighbouring countries Austria and Switzerland have very different reported patterns, in which Austria has a relatively high damage level for almost all years for both wind and SBB damage but relatively low peaks, while Switzerland has large storm damage followed by large SBB damage after the 1999 storm, but low or absent storm

damage except from that. The periods of zero bark beetle damage recorded indicate periods with low monitoring activity caused by low bark beetle activity. E.g., the country level compilation and gap filling of the DFDE database (Patacca et al., 2022) show seven times higher expert gap filled SBB damage than reported 1990-2005 for Sweden, and in France the machine learning gap filling resulted in 2.4 times higher damage 1991-2000.

Warm weather accompanied with drought are often seen as a contributing factor (e.g. Bakke, 1983), but has both

triggered and contributed to sustained outbreaks of SBB in Europe in recent years (Nardi et al., 2023; Trubin et al., 2022). There was no county-level data of SBB damage available for Sweden and France later than 2010, but country totals (Patacca et al., 2022) show levels that greatly exceed our small peak in modelled damage during 2018-2019. Drought has generally been seen as the main driver of these events (George et al., 2022; Kärvemo et





al., 2023). With the data used in our calibration and the additive form of the response function we might have
underestimated this response, and for Austria and Switzerland the model also underestimated the drought-induced
peak even when data for those years were included. Part of the reason for underestimating the damage could be
that LPJ-GUESS may be failing to simulate sufficiently increased water stress in the vegetation during this time
period. For instance, in Austria and Switzerland, there were no abnormally low values of the water stress scaler
wscal (indicating water stress) post 2015. This may be a failing of the model parameterization for water stress or
of the input climate forcing dataset (Steinkamp and Hickler, 2015). We note, however, that the 2018 drought
generates a fairly strong water stress response in the simulations for southern Sweden (Fig. 7) and this is reflected
in some increase in beetle damage in the years 2018 and 2019 (Fig. 3). An additional contributing factor that not
has been explicitly accounted for in the present study is root rot, which can significantly increase the severity of
an outbreak (Honkaniemi et al., 2018).

The simple test with a 2° C higher temperature gave an expected increase in modelled damage in S Sweden and
Austria. In Switzerland and NE France, however, a reduction was simulated, mainly related to a change of the
predisposing spruce biomass $>d_{lim}$ as simulated growth of the boreal-parameterized BNE PFT was reduced in the
warmer climate. The SBB phenology response also reaches a plateau when the climate is warm enough to allow
a complete second generation to emerge every year. A third generation can be completed with very hot conditions,
but the consequences for SBB population dynamics depends on host tree availability.

Whilst predicting absolute damage levels is challenging, our model is generally effective at indicating when there
are elevated periods of damage risk due to SBB (Fig. 7). The most obvious case is year 2003 with a strong drought
(Granier et al., 2007) that contributed to prolonged outbreaks in Switzerland, Austria and NE France, but also in
the recent years (2017-2019) the indicators also show increased risks coinciding with observed damage for all
assessed countries (Patacca et al., 2022). Combined with the process detail related to forest structure and
management, this means that the model can be a powerful tool to explore how different forest structures and
climate and management scenarios might interact to shift forests towards increased or decreased vulnerability.

The model that we have developed here is parameterised for SBB. However, many different species of bark beetles
that have the potential to cause large outbreaks and tree mortality, meaning that effectively accounting for the
impact of these species on large-scale forest dynamics requires that we can develop methods to generalise
responses across species to some degree. These other species of bark beetles can be specific in temperature sums
for evolving and dormancy periods and for density dependent defence overcome and competition (Bentz et al.,
2019). A common pattern for many species is increased risk for outbreaks related to drought (Reed and Hood,
2021), which is positive for a general concept of bark beetle damage modelling. The implementation of the SBB
outbreak dynamics in LPJ-GUESS has introduced a concept of insect functional types (IFTs), which can be
modelled in parallel with different response functions and specific PFT hosts. With this concept the model could
also be applied in other parts of the world, if sufficient data on bark beetle phenology and damage exist to
parameterise and calibrate the model.





## 6. Conclusions

Nature and human interactions drive and control SBB outbreaks in the intensively managed forest in Europe, which makes modelling challenging. The modelling concept we present here was able to catch the main timing and duration of observed damage but there was a substantial spread in absolute agreement, as also reflected in uncertainty in parameter estimates. With more detailed information of the human factors, mainly in form of salvage logging and sanitary cutting, it should be possible to improve the explanatory power of the model. The

SBB module is sensitive to the climate change signal, though the magnitude of temperature driven SBB damage seen in recent years was underestimated. In the modelling framework of the powerful LPJ-GUESS dynamical vegetation model it can, therefore, be a useful tool for exploring the SBB vulnerability of future climate and management scenarios.

### Code availability

The underlying LPJ-GUESS code subversion revision 13130 and simulation settings are archived at Zenodo (Lagergren et al., 2024b). Analysis code is available via Github (https://github.com/LPJ-GUESS/spruce_bark_beetle; doi on acceptance).

### Data availability

Model simulations underlying the results herein are archived at Zenodo (Lagergren et al., 2024a).

### 530 Author contribution

TP initiated the study. FL developed the SBB damage modelling concept with interaction from AMJ and TP. ML invented and implemented the insect functional type concept and coded the phenology of SBB into LPJ-GUESS. FL conducted all simulations and analysis of the results with feedback from all authors. FL lead the writing of the manuscript with contribution from all authors.

### 535 Competing interests

The authors declare that they have no conflict of interest.

### Acknowledgements

The model developments described in this paper have been funded under the European Union's Horizon Europe research and innovation programme (grant agreement numbers 101059888, CLIMB-Forest; 101056755, 540 ForestPaths; 101084481) and the Horizon 2020 programme (grant agreement no. 758873, TreeMort) as well as from the ForestValue programme, the European Commission, Vinnova, the Swedish Energy Agency and Formas for the project FORECO. This study is a contribution to the Swedish government's strategic research areas BECC



and MERGE and the Nature-based Future Solutions profile area at Lund University. We thank Arjan Meddens,
Rupert Seidl, Nikica Ogris and Cornelius Senf for discussions on the early stages of this work and Markus Kautz
for providing phenology data.

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
