# Peer review of "Combining empirical and mechanistic understanding of spruce bark beetle outbreak dynamics in the LPJ-GUESS (v4.1, r13130) vegetation model"

_Geoscientific Model Development, 2024_

## Referee Comment (RC2)

**Review: "Combining empirical and mechanistic understanding of spruce bark beetle outbreak dynamics in the LPJ-GUESS (v4.1, r13130) vegetation model" (gmd-2024-239)**

In this manuscript, the authors describe the implementation and performance of a module to represent spruce bark beetle infestations in the LPJ-GUESS dynamic global vegetation model (DGVM). This is important work because, while pests and disease are major drivers of forest disturbance in some regions, they are woefully underrepresented in DGVMs. I appreciate how the authors seemed to focus on building a system that is extensible for additional pest types, since the important species vary widely across the world.

The manuscript is for the most part well-written, but I do have some questions and suggestions for clarification. Similarly, the analyses are appropriate, and the figures are mostly clear. I thus recommend this to be reconsidered after what are probably minor revisions, but they are important enough that I would like to see them before they're accepted.

1. One citation of previous work is missing: Marie et al. (2024, *GMD*): "Simulating *Ips typographus* L. outbreak dynamics and their influence on carbon balance estimates with ORCHIDEE r8627"
2. L98: Patch area is unnecessary
3. L125-127: Not sure what this means
4. L128: What age classes? Are those in the inventory data you're talking about? Inventory data should probably get its own introductory paragraph, before you start talking about how you incorporate forest management into LPJ-GUESS sims.
5. L132: 5 patches per treatment seems low. Have you tested how replicable any of the results are at 5 patches, or the effect of increasing to 20 patches? It would be useful to demonstrate that 5 patches are enough for replicable results, or to increase the number of patches until it is.
6. L129-141:
   a. I would use "unmanaged vegetation" rather than PNV. There are still anthropogenic effects on the unmanaged patches (e.g., population density affecting fire).
   b. Were patch-destroying disturbances also turned off for unmanaged patches?
   c. Was fire turned off for any patches?
7. L153-166:
   a. Where does the initial (t=0) bark beetle population come from?
   b. Is there no term to describe how population increases when a low population experiences a big surge in substrate? Or is that 2nd term on the LHS of Eq. 4

positive at low values and negative at high values? (Later, in Fig. 1, I see that the latter is the case. But you should mention this before then.)

8. L168-169:
   a. What do you mean, ranges? Is this for parameterization purposes? Explain.
   b. What parts of your model correspond to which parts of the Marini et al. (2017) model?
   c. What are the ranges?
9. L171-172: Is that realistic?
10. Fig. 1a:
    a. Are the ranges used just the dark part? Or the light + dark parts?
    b. Where does the -3.8 number come from?
11. Fig. 1b-e:
    a. Lines are too thin, and pink is especially hard to see.
    b. What are the k parameters? They're not defined until after the figure. In the caption, refer the interested reader to eq. 7.
12. L187-192: Eqs. 7 and 8 were initially confusing because I couldn't figure out why Rgridcell wouldn't just be the mean of all Rpatch values. However, these aren't actually describing the population change exponent as is implied by the use of R; they're describing different additive terms *within the equation for* R. (On reread, I see that the Rgridcell and Rpatch convention is introduced in Fig. 1, but that's easy to miss.) For clarity, the right-hand side of these equations should be $f\left(P_{gridcell\ t-1}\right)$ and $f\left(P_{patch\ t-1}/L\right)$, respectively. In addition, before showing the equations, remind the reader in words what those terms are supposed to represent (respectively: effect of landscape-scale and substrate-scale competition [or the relief thereof, at low densities]).
13. It seems like the change in available material (L) is used in all these equations. I think that makes sense. But it also sounds from your text like only the POSITIVE component of L is used; e.g., L196-197. What about the NEGATIVE component— losses to fire, decomposition, and bark beetles? L should represent the NET change in substrate availability, no?
14. L202-203:
    a. Why was wscal calculated for both the previous and current year?
    b. This raises the question: When is this calculation happening? Is it at the end of the calendar year?
15. L205: I'm confused about how the weighted mean (wscal_mean) is calculated. Please add an equation explaining it.
16. L207: How is autumn swarming period defined? A reference to Marini et al. (2017) isn't enough; for *GMD* you need to go into these kinds of details.

17. L214-221:
    a. When salvage and/or sanitary cutting is performed, is the prescribed harvest fraction reduced for that year? E.g., if damage_available > salvmax, there should be no additional capacity for wood harvest in the first half of the year (when salvage/sanitary cutting is performed).
    b. Where did the 5 m3 number come from?
18. L252-271:
    a. It's not surprising that CRU wind data aren't very informative for wind damage, as mean wind speeds don't account for damaging gusts. For a similar finding with regard to fire Lasslop et al. (2015, DOI:10.1071/wf15052)—this would be interesting for you to note/cite.
    b. Wouldn't it be simpler to just force with the observed storm damage? I think you don't do this because you want to account for the cohort/height-specific situation. This should be mentioned.
19. L276-279: Is "CRU" here referring to the CRU-JRA dataset? If so, replace all bare "CRU" references with "CRU-JRA." If not, cite the CRU dataset separately.
20. Table 2: Include a column referring to the equation(s) where each parameter is used.
21. L333-336: I don't understand.

Results

22. L405-406: This is confusing. What is "the LPJ-GUESS run with calibrated parameters"? I thought that was what you were describing at the top of this paragraph.
23. Sect. 3.4 adds nothing; it can be deleted, with any important information moved into the Discussion.

Discussion

24. L422-423: What are galleries? What is "the defense?"
25. L425: "tree density" initially had me thinking in terms of individuals/ha. Rephrase to "wood density" for clarity. Unless… is it actually individuals/ha? If so, please explain the connection there.
26. L427-428: This is confusing. Each country/region had multiple gridcells, no?
27. L462-465: Not sure what this bit is adding to this paragraph.
28. L474: "but" doesn't seem to fit here.

29. L493-495: How do # of generations emerge from the phenology function, which seems to just be Eq. 14?

---

## Author Comment (AC1)

RC1: 'Comment on gmd-2024-239', Anonymous Referee #1

General comments:

This work aimed to develop a module for the simulation of spruce bark beetle impact in the vegetation model LPJ-GUESS. The developed model was calibrated and tested based on data from several European countries, producing plausible and meaningfully accurate results. While the developed model was proved applicable, I found a number of aspects that require clarification. My general comments are:

-   The used language is overly technical, and some paragraphs were not possible to understand (I listed some formal comments below). Revision is required to make the text accessible also to readers without deep technical understanding of the presented concept.

*Based on specific comments from both reviewers the text, especially in M&M, we will go through a significant revision that we think substantially will help accessibility*

-   The overall objective of this model development and its intended use were not clearly formulated, which resulted in combinations of simple empirical components with mechanistic components involving bark beetle population dynamics, phenology, etc. It was difficult to comprehend this logic – some implementation steps looked like workarounds helping to resolve technical problems.

*We will concentrate the end of the introduction to make the objective and philosophy clearer*

-   Some aspects of the development (e.g., salvaging implementation, wind impact on growing stock, etc., as in my comments below) seems to be rather arbitrary, lacking robust testing.

*Based on the more specific comments we will revise for clarity and explanation and motivation of implementations*

-   The discussion is vague, not addressing the limitations of the proposed model or critically confronting this development with other works.

*We will add comments on limitations and references to other works*

Specific comments:

-   Bark beetle sub-models have already been implemented in different models; however, the overview of these implementations has not been provided (or provided only partly). I suggest shortening parts of the introduction on disturbance development in Europe and go directly to the current situation in bark beetle modelling and limitations, which this study aims to address.

*We think that the background of spruce bark beetle description and damage pattern already is quite concise. We will add a wider discussion on the current status of modelling and limitations.*

-   The link between bark beetle ecology and population dynamics and implementation of these processes in the developed model was insufficiently documented, which hampers the assessment of this implementation. Bark beetle dynamics are driven by a number of processes (inciting, predisposing, amplifying, terminating the outbreak, etc.), which need to have their adequate counterparts (thought simplified) in the model (in L55 this complexity was somewhat narrowed to

phenology and forest conditions). In the current text, the information about beetle`s phenology; effects of windthrows, drought, salvage logging, etc. are rather scattered across the text, not providing a consistent framework to follow and understand the implementation of these processes. A section on Predisposing, triggering and contributing factors was placed on the end of the Results. However, this was done in a very inconsistent way, and the section consists of a single sentence only(?); therefore it is not very helpful.

*We will explain salvage and sanitary cutting in the introduction:* To further complicate the situation, counter measures such as salvage and sanitary cutting (SSC), which are effective in preventing and shorten outbreaks of SBB, have a high variation in intensity over time and space. Salvage logging of storm felled trees reduce the material where SBB can have very efficient regeneration and can even reduce the SBB population if the harvest is done between infection and emergence of the new generation. Sanitary cutting of infected living trees will take away the new generation, which can be a significant factor for ending an outbreak situation.

*The ending result section will be moved to the discussion.*

- The objectives (L85) say that the point is to … „catch(?) outbreak dynamics, utilize empirical relationships where available but make use of suitable mechanistic knowledge … ". This process resolution is very unclear, and it should be clearly defined with regard to the overall objective of this development. If the purpose is to simulate flexibly bark beetle dynamics under different management and climates, including, for example, the recent Europe-wide transitions from wind- to drought-driven dynamics, a high-level of process detail may be needed. If the point is to reproduce past dynamics and use the model in the range of past conditions, more empirical implementation could be OK. However, this objective and decisions about desired process complexity seem to be arbitrary (or insufficiently documented) in the paper. For example, Eq. 3 models bark beetle impact increase rate by the empirical model that used temperature, precipitation windfelled amounts, etc. as predictors. However, this empirical relationship is modified by the population-level processes such as negative feedback from denser beetle populations, swarming-induced competition. It would be very helpful to clarify in the beginning the degree of process complexity this model aims at and, if possible, keep it consistent.

*We will add a two sentences that describes what we aim for and how we will achieve it in the end of the introduction:* To achieve this, a semi-empirical SBB damage model, including SSC functionality, with components of negative feedback from a dense SBB population, amplification of damage after storm felling, where also warm and dry weather can trigger and contribute to sustained outbreaks, will be calibrated based on storm and SBB damage statistics from four countries in Europe. In this model, we seek to represent key aspects of the interaction of SBB with climate change and management, specifically phenology, SSC, interaction with storm and drought, whilst minimising process complexity by adopting more empirical elements wherever possible.

- Does this approach have any justification? „To enable that an outbreak can also be sustained at the highest population levels, the lowest possible total negative feedback from population size is just below the highest possible positive feedback from water stress and phenology (L….)". Does not it prevent outbreak termination due to internal regulation mechanisms? This approach sounds like a workaround rather than ecologically grounded solutions (but I admit I had a difficulty to understand this part).

*We will make substantial improvement on the description of the model and its components (e.g. see comments to referee #2 point 8b). We think that it will be clear that it can prevent outbreak termination with heat and drought, but in normal weather it will contribute to ending an outbreak.*

- I have a similar concern as in the previous point concerning the implementation of salvage logging (L215-220). This description sounds like a set of arbitrary decisions that technically allowed to include the effects of salvage logging into the model. However, no correspondence with a real dampening effect of salvage logging on outbreaks was presented.

*The decision to include salvage logging in the model was consistent with our aim to provide a model that could usefully be applied to explore climate-change and management interactions over continental scales. We will give a motivation for the setting of salvage logging:* In Sweden forest owners are not allowed to leave more than 5 m$^3$ of damaged spruce wood with d >10 cm per hectare after a storm, as regulated by the Swedish Forestry Act. *In 3.3 we clearly show the strong effect of salvage and sanitary cutting*.

- I am not sure if positive effect on water stress on beetle population growth can be termed as „positive feedback" (L…). Positive feedback typically has a different meaning. Concerning the term „positive feedback from phenology" – probably positive effect of temperature of bark beetle population growth (through altering phenology)?

*We will change to impact:* just below the highest possible positive impact of water stress and phenology

- What is the difference in R in Eq. 3 and R in Eq. 4? I suppose they represent different variables (Eq. 3 increase rate of forest volume loss, Eq. 4 forest damage by bark beetles?; as I inferred from the text), therefore, they should be represented differently. I could not understand how the two Rs from Eq. 3 and Eq. 4 interact.

*Good point, we will explain it:* As there is a linear dependency of *R* for *M* (Eq. 5) *R* representing increase rate of damage (Eq. 2) corresponds to *R* representing increase rate of $P_{patch}$.

- L185, Fig. 1 caption – it sounds a bit weird to use term "components" and "ranges of components". Cannot it be factors, variables, predictors, drivers or so?

*We think the terms are the right ones for what they stand for and we intend to still use them. We have made a lot of other changes to improve the figure text according to comments from referee #2.*

- „Shape of the functions for the components of the increase rate" Should not it be: Response of water stress coefficient driving bark beetle population growth to mean water stress? I generally found the used formulations very unnatural, making it difficult to understand the text .

*It stands after a reference to subfigures b-e so it does not refer to the water stress function. We will make this clearer by adding a reference of the different equations the subfigures are connected to in the end of the sentence: (b-e) Shape of the functions for the components of the increase rate of the bark beetle population index (R), b Eq. 7, c Eq. 8, d Eq. 10, e Eq. 11.*

- It is really extremely difficult to understand the logic of such statement: „As storms in Europe mainly occur in autumn and winter, and as the vegetation and bark beetle effect will be the same for a storm event in October or in February the next year, the storm damage statistics for a specific year were compiled for a storm season of July specified year until June the following year." Unfortunately, such formulations are frequent.

*We will revise the sentence to:* As storms in Europe mainly occur in autumn and winter, and the amount of damage caused by the storm is much more important than its timing during this period when considering its impact on bark beetle dynamics during the following growing season, the storm damage statistics for a specific year were compiled for a storm season of 12 months from July until June the next year when building the dataset used for the calibration.

- Section 2.4. I could not understand paragraph in L235 describing data availability. L239-240 – does this mean that there was only a single volume value for France and Austria (from 2008) and this single value was used to generate annual time series spanning 1961-2010? I m not convinced if this approach can be considered reliable. In the case of such data limitation, would not it be better to focus on countries with better data coverage?

*The storm damage data were available for each year, it was only the standing volume that was taken from one year. We state in the first paragraph of 2.4 that we used damage fraction for the model development.*

- L245 – the SDI concept from Patacca and its use in the current study would need to be much more elaborated. How is it to be used for evaluating the shift from wind to drought driven outbreaks in the developed simulation framework, as stated in L242?

*We admit that this statement was a bit too strong, we will revise the sentence to:* To test if this new situation in driving factors was important for the model parameterization, national level storm and bark beetle damage statistics from the Standardized Disturbance Index (SDI) dataset (Patacca et al., 2022) in years 2011-2019 for Switzerland and Austria were used.

- L255, wind implementation. As the text is whole overly complicated, I suggest simplifying this paragraph. It could be written directly that the wind impacts were prescribed to match the observed pattern, without elaborating on the experiment that did not work (driving the damage by real wind series data).

*We will simplify the first sentence of this paragraph to:* In order to focus on bark beetle outbreak dynamics without introducing additional uncertainties associated with wind data and wind damage modelling, we prescribed wind damage from observed data.

- L260-270. As far as I understand this part, the authors fixed a poor match of simulated wind damage with observations by introducing a correction by latitude, which correlated with productivity in Sweden (unpublished data?), and because there is higher damage in more productive sites, it should help simulate wind damage better. If this is correct, it looks like rather artificial solution for improving model performance. If the wind module is coupled with the vegetation model, should not the productivity and subsequently wind susceptibility be simulated as an emergent property? Without a need to imprint there this pattern externally.

*We will rewrite the rest of the paragraph for clarity and motivation for using the WL approach:* To still take advantage of the wind module's capacity to distribute wind damage among patches and cohorts' sensitivity (Eq. 1), a calibration was done to adjust WL so that modelled damage followed $DF_{storm}$ (denoted $WL_{stat}$). As a common linear scaling was used to go from $DF_{storm}$ to $WL_{stat}$, the exact $DF_{storm}$ time series will not be reproduced by the model. In a first step, a factor of 2 was found to approximately generate the same average level of $WL_{stat}$ calculated from $DF_{storm}$ as WL calculated from wind (see section 2.5 below) data for years 1990-2010. *The productivity data are part of the official statistics for Sweden but we have changed the weblink to a proper reference to the publication:* Roberge, C., Nilsson, P., Wikberg, P.-E., and Fridman, J.: Forest statistics 2023 - Official

Statistics of Sweden, Swedish University of Agricultural Sciences, Department of Forest Resource Management, Umeå, 168 pp.2023. *We will add a reference to the equation in the motivation to adjust $WL_{stat}$:* The scaling with LAT, as a proxy for productivity (LAT explains 82% of the variation in county average site quality class in Southern Sweden and 92% for all counties in Sweden (Tab. 3.11a, Roberge et al., 2023), is reasonable as a higher WL is needed to trigger a certain level of damage when the cohorts have a lower SI (as they have with lower productivity, see Eq. 1).

- L 289 and paragraph 295: This calibration procedure does not look rigorous and reproducible. First, the parameters were adjusted based on expert judgement. Second, only 8 out of fourteen parameters were calibrated because „we wanted to keep the range of response function …… as the original Marini model". I have doubts about such a sequence of arbitrary decisions.

*As a response to referee #2, the text describing this will be revised, we will add a reference to that section here: We will also add the motivation for excluding some of the parameters in the calibration:* These selected parameters were mainly related to the shape of the functions; other parameters were not included as we wanted to keep the range of the response functions at approximately the same magnitude as the weight of the original Marini et al. (2017) model (Eq. 3, see section 2.3). For this reason the max parameters where excluded from the calibration. As previously stated the $k_{base\_bm}$ parameter was set to a low value to avoid division by zero. To have a linear response within the wanted range (Fig 1a) of the $f$(phenology) function for the space of ASP (Fig. 1e) there was little room to adjust the parameters of Eq. 11 and they were therefore set fixed.

- L400 The switch between stand-alone Mtlab implementation and the „calibrated main base run" is confusing. Sounds more like developers' jargon than the text aimed at a broader audience

*We will revise and clarify the statement:* Up to this point all results are for the stand-alone Matlab implementation based on the vegetation from the default LPJ-GUESS run, i.e. with feedback from SBB damage from the default run instead of from the stand-alone.

- L403 – the 70 % overestimation for Austria was because of a single grid cell with missing negative feedback. The problem was fixed by removing this cell. In my opinion, this indicates a broader problem in the implementation, which should be fixed. Currently, it remains unclear how this missing negative feedback affected simulations in other remaining cells, where this effect could have been less pronounced than in the single Austrian cell.

*If it had been an issue in more cells it would have shown up in the results. The problem can only show up in the "stand-version" were there is no feedback to the vegetation, when the calibrated model is applied in LPJ-GUESS we got "normal" damage in the gridcell. But as the model was calibrated as having this high damage it gives an underestimation when applied. We will add an acknowledge statement regarding this problem:* It can, therefore, be concluded that this calibration process resulted in a calibrated model with conservative damage estimates.

- L410 – It is surprising that a 2° warming could cause such a severe spruce biomass reduction that it exerted a strong dampening effect on bark beetle damage (compensating for the amplifying effect on bark beetle activity). If this was the case, it would require exploring this vegetation feedback in greater detail. This issue was not addressed in Discussion; it just repeated the results.

*A comment to these results will be added:* as this PFT was close to its environmental limits and outside Norway spruce' native distribution in these regions (Caudullo et al., 2016), already in the present climate.

- Concerning the Discussion – the text much better written and clearer than the previous sections. However, the presented model implementation, its limitations and confrontations with other models were addressed only marginally. The discussion mostly described general aspects of bark beetle dynamics and modelling.

*We will try to put our study more in the context of earlier works. When discussing predisposing to SBB damage we will add:* In a remote-sensing based modelling study, tree height, soil moisture and nearby clear-cuts were together with high spruce volume identified as the most important factors predisposing forests to SBB damage (Müller et al., 2022). *We will add this in the discussion of difficulty to represent the high damage levels in the recent droughts:* Also for the most recent application of SBB damage with the ORCHIDEE vegetation model, a shortcoming when it comes to reflect last years' damage levels associated with extreme drought has been concluded (Marie et al., 2024). *And when discussing the +2-degree application we will add:* Similar tests resulting in a strong increase in modelled SBB damage has also been done by (Jönsson et al., 2012) and (Seidl and Rammer, 2017).

Technical corrections

Abstract requires revision. The introductory part on spruce bark beetle is overly long, while motivations for the presented development and the need for this solution are missing. Recommend avoiding terms salvage and sanitary felling in abstract, as their effect on bark beetle dynamics may not be clear to readers without forestry background. I did not notice in the results that the high variability of simulations was due to the variable effect of salvage logging (but I may have misunderstood this part).

*The first part of the introduction will be shortened to:* For evaluating the forests' performance in the future, dynamic vegetation models are important tools. Tree mortality is an important function in such models and, especially for needle leaved forest in the temperate and boreal zones, bark beetles are important for the mortality pattern. *And we will motivate the study by inserting:* with the aim to present a general concept that can be used also for other bark beetle species. *As we write "salvage logging of storm felled forest and sanitary cutting of infested trees" we think it is clear enough for the interested reader. The expression "large variability" was not good, we will change to:* there were discrepancies in levels, which partly can be related to salvage logging of storm felled forest and sanitary cutting of infested trees.

L55-60 not only empirical approaches exist, see, for example, implementation in iLand but also in other models

*We will add references to LandClim, iLand and ORCHIDEE:* Whilst landscape- and national-level models for SBB exist (De Bruijn et al., 2014; Jönsson et al., 2012; Marie et al., 2024; Seidl and Rammer, 2017; Seidl et al., 2014; Temperli et al., 2013), the capability to explicitly model historical forest damage from SBB has not yet been demonstrated at European-scale.

L160-165 negative feedback. The paragraph is not possible to understand, requires revision

*As the population index has a direct negative impact on the population index in the next modelling step we use the well-established "negative feedback" term. The paragraph has been revised for clarity in after more detailed comments from referee #2.*

L170 – this paragraph seems essential, but I did not manage to understand it. Suggest revising this entire section to make it understandable (and reproducible) also for a reader without deep technical understanding of this framework.

*The paragraph will go through a comprehensive revision:* The fraction of the different age and management classes, from the landcover functionality in LPJ-GUESS, were used to calculate Pgridcell weighted over the classes. With all variables in the (Marini et al., 2017) model at +/- 2 standard deviation from the mean, *R* has a range of -4.66 – 3.36, but interactions between variables prevent it to reach higher numbers. *R* calculated from the observation data used in the present study (see 2.4 below) has a range of -2.2 – 2.9, but initial high numbers in the start of an outbreak was often missing as inventories started first under an outbreak situation. The total range of *R* (Fig. 1a) in the presented model was set to -3.8 – 6.0, where the possible range of the different parts of the model (Eq. 4, Fig. 1a) were given weights of the similar magnitude as in the (Marini et al., 2017) model. The maximum *R* can also be translated to an extreme case of population increase rate with two successful generations in a year with 21 female offspring per mother ($e^{6.1}$ = 21.12). To enable that an outbreak can also be sustained at the highest population levels, the lowest possible total negative feedback from population size is just below the highest possible positive impact of water stress and phenology.

L198 - "that goes from zero to total shutdown of photosynthesis" – consider revising the language

*This section will be substantially revised:* The dependency of $Pg_{ridcell}$, $Pp_{atch}$ and *L* corresponds to $D_{storm}$ and $D_{SBB}$ in Marini et al. (2017) model. To take advantage of LPJ-GUESS' ability to model drought impact the Marini et al. (2017) dependency of rainfall was replaced with a dependency of the ratio between water supply to the canopy and canopy water demand (wscal), as calculated by LPJ-GUESS. The value goes from zero at complete shutdown of photosynthesis and transpiration to one at no stress, was used to assess the dependency of drought (1 - wscal, Fig. 1d, Eq. 4):

The authors operate across the text (already in abstract) with terms salvage logging and sanitary logging, and effect of these operations on bark beetle outbreaks. This concept can be unclear for the readers as these effects are not properly explained. Moreover, the definition of salvage (removal of windfelled trees) and sanitary (preventative removal of infested trees) is possible, but it is far from generally accepted and used definition.

*We will now explain the terms in the introduction:* To further complicate the situation, counter measures such as salvage and sanitary cutting (SSC), which are effective in preventing and shorten outbreaks of SBB, have a high variation in intensity over time and space. Salvage logging of storm felled trees reduce the material where SBB can have very efficient regeneration and can even reduce the SBB population if the harvest is done between infection and emergence of the new generation. Sanitary cutting of infected living trees will take away the new generation, which can be a significant factor for ending an outbreak situation.

L214 infected trees. Probably infested.

*We will change to infested.*

L322 The sentence is not possible to understand: "To test the robustness of the approach to test the model for different parameter combinations with structure and Lmort prescribed from an LPJ-GUESS simulation with default parameters, LPJ-GUESS was finally run with the optimized parameter set, with feedback of the damage associated with that setting to the simulated vegetation". Unfortunately, such cases are frequent across the text.

*We will reformulate to:* To test the robustness of the approach to calibrate the model for different parameter combinations with structure and $L_{mort}$ prescribed from an LPJ-GUESS simulation with default parameters, LPJ-GUESS was finally run with the optimized parameter set, which generated the right corresponding feedback of the damage associated with that setting to the simulated vegetation.

Unclear citation in L141 (Pugh et al. Manuscript)

*We will update the citation with the most recent status or replace it*

L331 "common model with the main base-run optimization" Consider please that readers main not have a deep technical understanding of this procedure

*We will reformulate to:* Most of the parameters of the model common for all four regions in the main base-run optimization (with SSC and not including calibration data for Switzerland and Austria 2011-2019)

L333 The same as above – "It should be noted that the calibration always was based on data from all countries, also when the optimum model for the regions countries was selected, which explains why there is a difference for S Sweden and NE France between Table 3 and Table S1a and between Table S1b and Table S1c." It is necessary to find a language that makes these results accessible to and reproducible by a broader community.

*Based also on comments from referee #2, we will reformulate to:* It should be noted that all calibrations were based on data from all countries, then the optimum model among the $7^7$-parameter space for the regions countries, or all together, was selected, which explains why there is a difference for S Sweden and NE France when including calibration data 2011-2019 for Austria and Switzerland (Table 3 vs Table S1a and Table S1b vs Table S1c).

In this manuscript, the authors describe the implementation and performance of a module to represent spruce bark beetle infestations in the LPJ-GUESS dynamic global vegetation model (DGVM). This is important work because, while pests and disease are major drivers of forest disturbance in some regions, they are woefully underrepresented in DGVMs. I appreciate how the authors seemed to focus on building a system that is extensible for additional pest types, since the important species vary widely across the world.

The manuscript is for the most part well-written, but I do have some questions and suggestions for clarification. Similarly, the analyses are appropriate, and the figures are mostly clear. I thus recommend this to be reconsidered after what are probably minor revisions, but they are important enough that I would like to see them before they're accepted.

1. One citation of previous work is missing: Marie et al. (2024, GMD): "Simulating *Ips typographus* L. outbreak dynamics and their influence on carbon balance estimates with ORCHIDEE r8627"

*We will add references to the LandClim, iLand and ORCHIDEE models and rephrased the sentence referring to them to make it more specific:* Whilst landscape- and national-level models for SBB exist (De Bruijn et al., 2014; Jönsson et al., 2012; Marie et al., 2024; Seidl and Rammer, 2017; Seidl et al., 2014; Temperli et al., 2013), the capability to explicitly model historical forest damage from SBB has not yet been demonstrated at European-scale. *And we will also explain more details of what is missing:* The dynamical vegetation model (DVM) ORCHIDEE has recently been updated with a mechanistic SBB functionality (Marie et al., 2024), but lacks SSC functionality. The iLand landscape simulator has mechanistic SBB components and can simulate salvage logging, but has been evaluated in protected areas to rule out the influence of SSC (Seidl and Rammer, 2017).

2. L98: Patch area is unnecessary

*It will be removed.*

3. L125-127: Not sure what this means

*We will make the sentence clearer:* The age-class data had a regional resolution for France and a national for Austria and Switzerland. For Sweden national inventory data for 2008-2012 with a county resolution were used instead of the Poulter et al. (2019) data

4. L128: What age classes? Are those in the inventory data you're talking about? Inventory data should probably get its own introductory paragraph, before you start talking about how you incorporate forest management into LPJ-GUESS sims.

*We still think it can be placed where it is but we admit that it was poorly expressed and will change the entire sentence:* To get also weights for CCF and potential natural vegetation (PNV) from the datasets, the short and long CCF classes were used to represent the fraction of the 91-110 and 111-140 years age classes in the inventory data respectively, and PNV was used to represent the fraction of forest older than 140 years.

5. L132: 5 patches per treatment seems low. Have you tested how replicable any of the results are at 5 patches, or the effect of increasing to 20 patches? It would be useful to demonstrate that 5 patches are enough for replicable results, or to increase the number of patches until it is.

*Five patches may be a low number for PNV where random disturbances occur but it has a quite low total fraction. For the managed patches the random disturbances are turned off after management is initiated and there are only very minor differences, mainly in soil carbon, due to the different disturbance histories. Also, since the calibration is carried out at region/country level, the total number of gridcells involved is quite high. So, the effective number of replicates is high? So, we would agree with the reviewer if the calibration/evaluation was made at gridcell level, but it is not a problem at the region/country level. We do not intend to revise this.*

6. L129-141:

a. I would use "unmanaged vegetation" rather than PNV. There are still anthropogenic effects on the unmanaged patches (e.g., population density affecting fire).

*This is true, although the mentioned link of fire with population density is in fact the only such effect. However, PNV is a well-established term that has been used in many LPJ-GUESS publications in recent years. "Unmanaged vegetation" can have a temporary nature and is not crystal clear either. On this basis we prefer to keep the term PNV.*

b. Were patch-destroying disturbances also turned off for unmanaged patches?

*No, we point the reviewer to the formulation "after the introduction of management in a patch, these were turned off".*

c. Was fire turned off for any patches?

*This information will be added with a new sentence:* Fire disturbance was simulated with the BLAZE module and it was also turned off for managed patches.

7. L153-166:

a. Where does the initial (t=0) bark beetle population come from?

*This information will be added:* At start of a simulation $P_{patch}$ was initiated with a value of 10 for all patches.

b. Is there no term to describe how population increases when a low population experiences a big surge in substrate? Or is that 2$^{nd}$ term on the LHS of Eq. 4 positive at low values and negative at high values? (Later, in Fig. 1, I see that the latter is the case. But you should mention this before then.)

*We will make it clearer at this point (though it is later explained in conjunction with Eq. 9) by adding a comment:* relative to the amount of substrate with no defence (*L*, typically with a high value after storm damage)

8. L168-169:

a. What do you mean, ranges? Is this for parameterization purposes? Explain.

*We now refer to equation and figure and specify by "outcome range":* the possible outcome range of the different parts of the model (Eq. 4, Fig. 1a) were given weights

b. What parts of your model correspond to which parts of the Marini et al. (2017) model?

*We will add:* The dependency of $P_{gridcell}$, $P_{patch}$ and $L$ corresponds to $D_{storm}$ and $D_{SBB}$ in Marini et al. (2017) model. To take advantage of LPJ-GUESS' ability to model drought impact the Marini et al. (2017) dependency of rainfall was replaced with a dependency of the ratio between water supply to the canopy and canopy water demand (wscal) *The temperature part is already explained but the formulation will be revised to:* For a more mechanistic approach of taking phenology into account, the dependency of $T$ in the (Marini et al., 2017) model was replaced with a dependency on the length of the autumn swarming period (ASP) in comparison with the grid-cell specific 30-year average as

c. What are the ranges?

*We now try to explain and motivate our settings more clearly:* With all variables in the (Marini et al., 2017) model at +/- 2 standard deviation from the mean, $R$ has a range of -4.66 – 3.36, but interactions between variables prevent it to reach higher numbers. $R$ calculated from the observation data used in the present study (see 2.4 below) has a range of -2.2 – 2.9, but initial high numbers in the start of an outbreak were often missing as inventories only began when already under an outbreak situation. The total range of $R$ (Fig. 1a) in the presented model was set to -3.8 – 6.0, where the possible range of the different parts of the model (Eq. 4, Fig. 1a) were given weights of the similar magnitude as in the (Marini et al., 2017) model. The maximum R can also be translated to an extreme case of population increase rate with two successful generations in a year with 21 female offspring per mother ($e^{6.1} = 21.12$).

9. L171-172: Is that realistic?

We now acknowledge this as an extreme case: *The maximum R can also be translated to an extreme case of population increase rate*

10. Fig. 1a:

a. Are the ranges used just the dark part? Or the light + dark parts?

*We will revise the description for clarity:* For $f(P_{gridcell})$ and $f(P_{patch} / L)$, the light shaded areas show the part of the ranges that were varied in the parameter optimization, where the sum of the minimum of $f(P_{gridcell})$ and $f(P_{patch} / L)$ were kept constant to have the possible total negative feedback from the population index constant (see section 2.6).

b. Where does the -3.8 number come from?

*We will refer to section 2.6 where it is explained:* (see section 2.6)

11. Fig. 1b-e:

a. Lines are too thin, and pink is especially hard to see.

*We will revise line thickness and take a darker colour of pink:*

[Figure]

b. What are the k parameters? They're not defined until after the figure. In the caption, refer the interested reader to eq. 7.

*We now refer to all the equations and Table 2 and Section 2.6, where the k parameters are explained: (b-e) Shape of the functions for the components of the increase rate of the bark beetle population index (R), b Eq. 7, c Eq. 8, d Eq. 10, e Eq. 11. The default parameter setting (Table 7) is shown by thick grey lines (b-d). The functions are also shown in colour for the min and max value of parameters included in the calibration and sensitivity analysis (section 2.6), using the default setting for the other parameters.*

12. L187-192: Eqs. 7 and 8 were initially confusing because I couldn't figure out why Rgridcell wouldn't just be the mean of all Rpatch values. However, these aren't actually describing the population change exponent as is implied by the use of R; they're describing different additive terms within the equation for R. (On reread, I see that the Rgridcell and Rpatch convention is introduced in Fig. 1, but that's easy to miss.) For clarity, the right-hand side of these equations should be $f(P_{gridcell\ t-1})$ and $f(P_{patch\ t-1})/L)$, respectively. In addition, before showing the equations, remind the reader in words what those terms are supposed to represent (respectively: effect of landscape-scale and substrate-scale competition [or the relief thereof, at low densities]).

*All the equations describing the components of Eq. 4 will get their left-hand side (we assume that the reviewer has mixed left and right) replaced with the terms in Eq. 4 for consistency with that equation and Figure 1 (Eq. 7, 8, 10 and 11). We will also refer to Eq. 4 when the equations are presented.*

13. It seems like the change in available material (L) is used in all these equations. I think that makes sense. But it also sounds from your text like only the POSITIVE component of L is used; e.g., L196-

197. What about the NEGATIVE component—losses to fire, decomposition, and bark beetles? L should represent the NET change in substrate availability, no?

*L is not the net change but the absolute amount, which is stated in the text directly after Eq. 9.*

14. L202-203:

a. Why was wscal calculated for both the previous and current year?

*As it has been shown before that the previous year can be as important. We will change the formulation to:* The mean wscal calculated over the month May-July for both previous year ($wscal_{t-1}$) and current year ($wscal_t$) for the BNE PFT were used. Based on the (Marini et al., 2017) model, the data from the previous year were given a three times higher weight in the default setting, but in the calibration and sensitivity analysis (see section 2.4 below) the previous year weight (kpyw) was varied between ¼ to 4:

b. This raises the question: When is this calculation happening? Is it at the end of the calendar year?

*This is important information that we have missed to include, thank you, we will add this paragraph in the end of section 2.3:* The bark beetle accounting and application of damage is placed in the "mortality_guess" function in the vegdynam.cpp code together with the wind damage application, and it is called at the end of each simulated year (Lagergren et al., 2024a).

15. L205: I'm confused about how the weighted mean (wscal_mean) is calculated. Please add an equation explaining it.

*An equation will be added:* $wscal_{mean} = \dfrac{wscal_{t-1}k_{pyw} + wscal_t}{k_{pyw} + 1}$

16. L207: How is autumn swarming period defined? A reference to Marini et al. (2017) isn't enough; for GMD you need to go into these kinds of details.

*An explanation will be added:* where ASP is the number of flight days of the first new generation according to Jönsson et al. (2011).

17. L214-221:

a. When salvage and/or sanitary cutting is performed, is the prescribed harvest fraction reduced for that year? E.g., if damage_available > salvmax, there should be no additional capacity for wood harvest in the first half of the year (when salvage/sanitary cutting is performed).

*As carbon fluxes were not the focus of the present study, to simplify the simulation setup the salvage and salvage logging and sanitary cutting was not applied in the main vegetation accounting of the model and did not interfere with the prescribed harvest. We only applied it within the bark beetle outbreak calculations. We will explain this:* At this stage of the model development the effect of the salvage and sanitary cutting were just applied in the bark beetle accounting, the damaged trees were not removed in the main carbon accounting of the model. This can cause some underestimation of the heterotrophic respiration, but it was considered as insignificant for the present study which only focuses on SBB outbreak dynamics.

b. Where did the 5 m3 number come from?

*An explanation will be added:* In Sweden forest owners are not allowed to leave more than 5 m3 of damaged spruce wood with d >10 cm per hectare after a storm, as regulated by the

Swedish Forestry Act (Swedish Forest Agency, https://www.skogsstyrelsen.se/en/laws-and-regulations/skogsvardslagen/, last access: 12 May 2025).

18. L252-271:

a. It's not surprising that CRU wind data aren't very informative for wind damage, as mean wind speeds don't account for damaging gusts. For a similar finding with regard to fire Lasslop et al. (2015, DOI:10.1071/wf15052)—this would be interesting for you to note/cite.

*As a response to Reviewer 1, we have instead simplified the text.*

b. Wouldn't it be simpler to just force with the observed storm damage? I think you don't do this because you want to account for the cohort/height-specific situation. This should be mentioned.

*You are completely right, which we now explain:* In order to focus on bark beetle outbreak dynamics without introducing additional uncertainties associated with wind data and wind damage modelling, we prescribed wind damage from observed data. To still take advantage of the wind module's capacity to distribute wind damage among patches and cohorts' sensitivity, a calibration was done to adjust wind load (Eq. (6) in Lagergren et al., 2012) so that the observed damage was reached.

19. L276-279: Is "CRU" here referring to the CRU-JRA dataset? If so, replace all bare "CRU" references with "CRU-JRA." If not, cite the CRU dataset separately.

*All bare "CRU" references will be replaced with "CRU-JRA".*

20. Table 2: Include a column referring to the equation(s) where each parameter is used.

*The column will be added.*

21. L333-336: I don't understand.

*We have rewritten this section:* It should be noted that all calibrations were based on data from all countries, then the optimum model among the $7^7$-parameter space for the regions countries, or all together, was selected, which explains why there is a difference for S Sweden and NE France when including calibration data 2011-2019 for Austria and Switzerland (Table 3 vs Table S1a and Table S1b vs Table S1c).

Results

22. L405-406: This is confusing. What is "the LPJ-GUESS run with calibrated parameters"? I thought that was what you were describing at the top of this paragraph.

*This is described in section 2.7 for which we will edit the header to reflect that this test is described there:* 2.7. Robustness test and exploration of the climate change signal

23. Sect. 3.4 adds nothing; it can be deleted, with any important information moved into the Discussion.

*We will move this part to the beginning of the discussion where it fits better*

Discussion

24. L422-423: What are galleries? What is "the defense?"

*We think that most readers are familiar with the gallery term but for clarity we have revised to:* the beetles need a dense cover of beetle larvae galleries. *We have specified to:* resin defence

25. L425: "tree density" initially had me thinking in terms of individuals/ha. Rephrase to "wood density" for clarity. Unless… is it actually individuals/ha? If so, please explain the connection there.

*It is indeed trees per hectare, we will make this clear and explain the connection:* The tree density (trees per ha) determines how quickly a tree reaches $d_{lim}$, as reduced density means increased diameter growth, and, in turn, depends on plant number, thinning and mortality.

26. L427-428: This is confusing. Each country/region had multiple gridcells, no?

*As is shown in Fig. 2, there were multiple gridcells. We will try to make the sentence less confusing by not using the "one point in time" phrase:* the size of age classes was determined for one occasion at country or county scale

27. L462-465: Not sure what this bit is adding to this paragraph.

*We will keep it but revise to make it clearer what is challenging for dynamic forest models:* Many SBB models require an initial population or damage level, which means that they work in relative terms (Marini et al., 2017; Soukhovolsky et al., 2022), but a dynamic forest model needs to operate with absolute damage levels making the modelling more challenging.

28. L474: "but" doesn't seem to fit here.

*We will revise to:* but in recent year it has also triggered as well as contributed to sustained outbreaks of SBB in Europe (Nardi et al., 2023; Trubin et al., 2022).

29. L493-495: How do # of generations emerge from the phenology function, which seems to just be Eq. 14?

*We will motivate the use of ASP instead of number of generations in connection to Eq. 11:* The dependency of ASP instead of, e.g., the number of generations per year was chosen as ASP is a continuous variable which better catch the average when there is a variability in the climate, such in mountainous regions, then a discrete variable. *We will also acknowledge and remind the readers that we use ASP and not number of generations:* The SBB phenology response, which is a function of the length of the first generation's swarming period, also reaches a plateau when the climate is warm enough to allow a complete second generation to emerge every year.

---

## Author Response (AR1)

Our comments are written *in italic red font*. Revised text is written in plain red text, for which we give reference to the beginning line number of the revised annotated version. Note also that we have used EndNote for handling references and that changes to those do not show up as revised.

RC1: 'Comment on gmd-2024-239', Anonymous Referee #1

**General comments:**

This work aimed to develop a module for the simulation of spruce bark beetle impact in the vegetation model LPJ-GUESS. The developed model was calibrated and tested based on data from several European countries, producing plausible and meaningfully accurate results. While the developed model was proved applicable, I found a number of aspects that require clarification. My general comments are:

- The used language is overly technical, and some paragraphs were not possible to understand (I listed some formal comments below). Revision is required to make the text accessible also to readers without deep technical understanding of the presented concept.

Based on specific comments from both reviewers the text, especially in M&M, has gone through a significant revision that we think substantially will help accessibility

- The overall objective of this model development and its intended use were not clearly formulated, which resulted in combinations of simple empirical components with mechanistic components involving bark beetle population dynamics, phenology, etc. It was difficult to comprehend this logic – some implementation steps looked like workarounds helping to resolve technical problems.

To make the objective and philosophy clearer we have concentrated the end of the introduction (L108)

- Some aspects of the development (e.g., salvaging implementation, wind impact on growing stock, etc., as in my comments below) seems to be rather arbitrary, lacking robust testing.

Based on the more specific comments from both reviewer we have made a substantial revision for clarity regarding explanation and motivation of the implementations

- The discussion is vague, not addressing the limitations of the proposed model or critically confronting this development with other works.

We have added comments on limitations and references to other works, se more specific comments below

**Specific comments:**

- Bark beetle sub-models have already been implemented in different models; however, the overview of these implementations has not been provided (or provided only partly). I suggest shortening parts of the introduction on disturbance development in Europe and go directly to the current situation in bark beetle modelling and limitations, which this study aims to address.

We think that the background of spruce bark beetle description and damage pattern already is quite concise. We have added a section on the current status of modelling and limitations (L72): The dynamical vegetation model (DVM) ORCHIDEE has recently been updated with a mechanistic SBB functionality (Marie et al., 2024), but lacks SSC functionality. The iLand landscape simulator has mechanistic SBB components and can simulate salvage logging, but has been evaluated in protected areas to rule out the influence of SSC (Seidl and Rammer, 2017).

- The link between bark beetle ecology and population dynamics and implementation of these processes in the developed model was insufficiently documented, which hampers the assessment of this implementation. Bark beetle dynamics are driven by a number of processes (inciting, predisposing, amplifying, terminating the outbreak, etc.), which need to have their adequate counterparts (thought simplified) in the model (in L55 this complexity was somewhat narrowed to phenology and forest conditions). In the current text, the information about beetle's phenology; effects of windthrows, drought, salvage logging, etc. are rather scattered across the text, not providing a consistent framework to follow and understand the implementation of these processes. A section on Predisposing, triggering and contributing factors was placed on the end of the Results. However, this was done in a very inconsistent way, and the section consists of a single sentence only(?); therefore it is not very helpful.

We now explain salvage and sanitary cutting in the introduction (L52): To further complicate the situation, counter measures such as salvage and sanitary cutting (SSC), which are effective in preventing and shorten outbreaks of SBB, have a high variation in intensity over time and space. Salvage logging of storm felled trees reduce the material where SBB can have very efficient regeneration and can even reduce the SBB population if the harvest is done between infection and emergence of the new generation. Sanitary cutting of infected living trees will take away the new generation, which can be a significant factor for ending an outbreak situation.

**The ending result section has been moved to the discussion (L490).**

- The objectives (L85) say that the point is to ... "catch(?) outbreak dynamics, utilize empirical relationships where available but make use of suitable mechanistic knowledge ... ". This process resolution is very unclear, and it should be clearly defined with regard to the overall objective of this development. If the purpose is to simulate flexibly bark beetle dynamics under different management and climates, including, for example, the recent Europe-wide transitions from wind- to drought-driven dynamics, a high-level of process detail may be needed. If the point is to reproduce past dynamics and use the model in the range of past conditions, more empirical implementation could be OK. However, this objective and decisions about desired process complexity seem to be arbitrary (or insufficiently documented) in the paper. For example, Eq. 3 models bark beetle impact increase rate by the empirical model that used temperature, precipitation windfelled amounts, etc. as predictors. However, this empirical relationship is modified by the population-level processes such as negative feedback from denser beetle populations, swarming-induced competition. It would be very helpful to clarify in the beginning the degree of process complexity this model aims at and, if possible, keep it consistent.

We have added two sentences that describes what we aim for and how we will achieve it in the end of the introduction (L108): To achieve this, a semi-empirical SBB damage model, including SSC functionality, with components of negative feedback from a dense SBB population, amplification of damage after storm felling, where also warm and dry weather can trigger and contribute to sustained outbreaks, will be calibrated based on storm and SBB damage statistics from four countries in Europe. In this model, we seek to represent key aspects of the interaction of SBB with

climate change and management, specifically phenology, SSC, interaction with storm and drought, whilst minimising process complexity by adopting more empirical elements wherever possible.

- Does this approach have any justification? "To enable that an outbreak can also be sustained at the highest population levels, the lowest possible total negative feedback from population size is just below the highest possible positive feedback from water stress and phenology (L....)". Does not it prevent outbreak termination due to internal regulation mechanisms? This approach sounds like a workaround rather than ecologically grounded solutions (but I admit I had a difficulty to understand this part).

We have made a substantial improvement on the description of the model and its components (e.g. see comments to referee #2 point 8b). We think that it will be clear that it can prevent outbreak termination with heat and drought, but in normal weather it will contribute to ending an outbreak.

- I have a similar concern as in the previous point concerning the implementation of salvage logging (L215-220). This description sounds like a set of arbitrary decisions that technically allowed to include the effects of salvage logging into the model. However, no correspondence with a real dampening effect of salvage logging on outbreaks was presented.

The decision to include salvage logging in the model was consistent with our aim to provide a model that could usefully be applied to explore climate-change and management interactions over continental scales. We now give a motivation for the setting of salvage logging (L263): In Sweden forest owners are not allowed to leave more than 5 m³ of damaged spruce wood with d >10 cm per hectare after a storm, as regulated by the Swedish Forestry Act. In in the results (section 3.3) we clearly show the strong effect of salvage and sanitary cutting.

- I am not sure if positive effect on water stress on beetle population growth can be termed as "positive feedback" (L...). Positive feedback typically has a different meaning. Concerning the term "positive feedback from phenology" – probably positive effect of temperature of bark beetle population growth (through altering phenology)?

We have changed to impact (L206): just below the highest possible positive impact of water stress and phenology

- What is the difference in R in Eq. 3 and R in Eq. 4? I suppose they represent different variables (Eq. 3 increase rate of forest volume loss, Eq. 4 forest damage by bark beetles?; as I inferred from the text), therefore, they should be represented differently. I could not understand how the two Rs from Eq. 3 and Eq. 4 interact.

Good point, we now explain it (L184): As there is a linear dependency of R for M (Eq. 5), R representing increase rate of damage (Eq. 2) corresponds to R representing increase rate of  $P_{patch}$ .

- L185, Fig. 1 caption – it sounds a bit weird to use term "components" and "ranges of components". Cannot it be factors, variables, predictors, drivers or so?

We think the terms are the right ones for what they stand for and we decided to still use them. We have made a lot of other changes to improve the figure text according to comments from referee #2.

- "Shape of the functions for the components of the increase rate" Should not it be: Response of water stress coefficient driving bark beetle population growth to mean water stress? I generally found the used formulations very unnatural, making it difficult to understand the text.

It stands after a reference to subfigures b-e so it does not refer to the water stress function. We have made this clearer by adding a reference to the different equations the subfigures are connected to in the end of the sentence (L215): (b-e) Shape of the functions for the components of the increase rate of the bark beetle population index (R), b Eq. 7, c Eq. 8, d Eq. 10, e Eq. 12.

- It is really extremely difficult to understand the logic of such statement: "As storms in Europe mainly occur in autumn and winter, and as the vegetation and bark beetle effect will be the same for a storm event in October or in February the next year, the storm damage statistics for a specific year were compiled for a storm season of July specified year until June the following year." Unfortunately, such formulations are frequent.

We have revised the sentence to (L286): As storms in Europe mainly occur in autumn and winter, and the amount of damage caused by the storm is much more important than its timing during this period when considering its impact on bark beetle dynamics during the following growing season, the storm damage statistics for a specific year were compiled for a storm season of 12 months from July until June the next year when building the dataset used for the calibration.

- Section 2.4. I could not understand paragraph in L235 describing data availability. L239-240 – does this mean that there was only a single volume value for France and Austria (from 2008) and this single value was used to generate annual time series spanning 1961-2010? I m not convinced if this approach can be considered reliable. In the case of such data limitation, would not it be better to focus on countries with better data coverage?

The storm damage data were available for each year, it was only the standing volume that was taken from one year. We state in the first paragraph of 2.4 that we used damage fraction for the model development.

- L245 – the SDI concept from Patacca and its use in the current study would need to be much more elaborated. How is it to be used for evaluating the shift from wind to drought driven outbreaks in the developed simulation framework, as stated in L242?

We admit that this statement was a bit too strong, and we have revised the sentence to (L302): To test if this new situation in driving factors was important for the model parameterization, national level storm and bark beetle damage statistics from the Standardized Disturbance Index (SDI) dataset (Patacca et al., 2022) in years 2011-2019 for Switzerland and Austria were used.

- L255, wind implementation. As the text is whole overly complicated, I suggest simplifying this paragraph. It could be written directly that the wind impacts were prescribed to match the observed pattern, without elaborating on the experiment that did not work (driving the damage by real wind series data).

We have simplified the first sentence of this paragraph to (L315): In order to focus on bark beetle outbreak dynamics without introducing additional uncertainties associated with wind data and wind damage modelling, we prescribed wind damage from observed data.

- L260-270. As far as I understand this part, the authors fixed a poor match of simulated wind damage with observations by introducing a correction by latitude, which correlated with productivity in Sweden (unpublished data?), and because there is higher damage in more productive sites, it should help simulate wind damage better. If this is correct, it looks like rather artificial solution for improving model performance. If the wind module is coupled with the vegetation model, should not the productivity and subsequently wind susceptibility be simulated as an emergent property? Without a need to imprint there this pattern externally.

We have rewritten the rest of the paragraph for clarity and motivation for using the WL approach (L317): To still take advantage of the wind module's capacity to distribute wind damage among patches and cohorts' sensitivity (Eq. 1), a calibration was done to adjust WL so that modelled damage followed DFstorm (denoted WLstat). As a common linear scaling was used to go from DFstorm to WLstat, the exact DFstorm time series will not be reproduced by the model. In a first step, a factor of 2 was found to approximately generate the same average level of WLstat calculated from DFstorm as WL calculated from wind (see section 2.5 below) data for years 1990-2010. The productivity data are part of the official statistics for Sweden but we have changed the weblink to a proper reference to the publication (L334): Roberge, C., Nilsson, P., Wikberg, P.-E., and Fridman, J.: Forest statistics 2023 - Official Statistics of Sweden, Swedish University of Agricultural Sciences, Department of Forest Resource Management, Umeå, 168 pp.2023. We have added a reference to the equation in the motivation to adjust WLstat (L331): The scaling with LAT, as a proxy for productivity (LAT explains 82% of the variation in county average site quality class in Southern Sweden and 92% for all counties in Sweden (Tab. 3.11a in Roberge et al., 2023), is reasonable as a higher WL is needed to trigger a certain level of damage when the cohorts have a lower SI (as they have with lower productivity, see Eq. 1).

- L 289 and paragraph 295: This calibration procedure does not look rigorous and reproducible. First, the parameters were adjusted based on expert judgement. Second, only 8 out of fourteen parameters were calibrated because "we wanted to keep the range of response function ...... as the original Marini model". I have doubts about such a sequence of arbitrary decisions.

As a response to referee #2, the text describing this has been revised, we have add a reference to that section here: We have also added the motivation for excluding some of the parameters in the calibration (L359): These selected parameters were mainly related to the shape of the functions; other parameters were not included as we wanted to keep the range of the response functions at approximately the same magnitude as the weight of the original Marini et al. (2017) model (Eq. 3, see section 2.3). For this reason the max parameters where excluded from the calibration. As previously stated, the  $k_{\text{base\_bm}}$  parameter was set to a low value to avoid division by zero. To have a linear response within the wanted range (Fig 1a) of the f(phenology) function for the space of ASP (Fig. 1e) there was little room to adjust the parameters of Eq. 12 and they were therefore set fixed.

- L400 The switch between stand-alone Mtlab implementation and the "calibrated main base run" is confusing. Sounds more like developers' jargon than the text aimed at a broader audience

We have revised and clarified the statement (L470): Up to this point all results are for the standalone Matlab implementation based on the vegetation from the default LPJ-GUESS run, i.e. with feedback from SBB damage from the default run instead of from the stand-alone.

- L403 – the 70 % overestimation for Austria was because of a single grid cell with missing negative feedback. The problem was fixed by removing this cell. In my opinion, this indicates a broader problem in the implementation, which should be fixed. Currently, it remains unclear how this missing negative feedback affected simulations in other remaining cells, where this effect could have been less pronounced than in the single Austrian cell.

If it had been an issue in more cells it would have shown up in the results. The problem can only show up in the "stand-alone-version" were there is no feedback to the vegetation, when the calibrated model is applied in LPJ-GUESS we got "normal" damage in the gridcell. But as the model was calibrated as having this high damage it gives an underestimation when applied. We have added an

acknowledge statement regarding this problem (L478): It can, therefore, be concluded that this calibration process resulted in a calibrated model with conservative damage estimates.

- L410 – It is surprising that a 2° warming could cause such a severe spruce biomass reduction that it exerted a strong dampening effect on bark beetle damage (compensating for the amplifying effect on bark beetle activity). If this was the case, it would require exploring this vegetation feedback in greater detail. This issue was not addressed in Discussion; it just repeated the results.

A comment to these results was added (L482): as this PFT was close to its environmental limits and outside Norway spruce' native distribution in these regions (Caudullo et al., 2016), already in the present climate.

- Concerning the Discussion – the text much better written and clearer than the previous sections. However, the presented model implementation, its limitations and confrontations with other models were addressed only marginally. The discussion mostly described general aspects of bark beetle dynamics and modelling.

We have tried to put our study more in the context of earlier works. When discussing predisposing to SBB damage we now add (L493): In a remote-sensing based modelling study, tree height, soil moisture and nearby clear-cuts were together with high spruce volume identified as the most important factors predisposing forests to SBB damage (Müller et al., 2022). We add this in the discussion of difficulty to represent the high damage levels in the recent droughts (L567): Also for the most recent application of SBB damage with the ORCHIDEE vegetation model, a shortcoming when it comes to reflect last years' damage levels associated with extreme drought has been concluded (Marie et al., 2024). And when discussing the +2-degree application we now add (L575): Similar tests resulting in a strong increase in modelled SBB damage has also been done by (Jönsson et al., 2012) and (Seidl and Rammer, 2017).

**Technical corrections**

Abstract requires revision. The introductory part on spruce bark beetle is overly long, while motivations for the presented development and the need for this solution are missing. Recommend avoiding terms salvage and sanitary felling in abstract, as their effect on bark beetle dynamics may not be clear to readers without forestry background. I did not notice in the results that the high variability of simulations was due to the variable effect of salvage logging (but I may have misunderstood this part).

The first part of the introduction text in the abstract has been shortened to (L9): For evaluating the forests' performance in the future, dynamic vegetation models are important tools. Tree mortality is an important function in such models and, especially for needle leaved forest in the temperate and boreal zones, bark beetles are important for the mortality pattern. And we now motivate the study by inserting (L19): with the aim to present a general concept that can be used also for other bark beetle species. As we write "salvage logging of storm felled forest and sanitary cutting of infested trees" we think it is clear enough for the interested reader. The expression "large variability" was not good, we have changed it to (L24): there were discrepancies in levels, which partly can be related to salvage logging of storm felled forest and sanitary cutting of infested trees.

L55-60 not only empirical approaches exist, see, for example, implementation in iLand but also in other models

We have added references to LandClim, iLand and ORCHIDEE (L57): Whilst landscape- and national-level models for SBB exist (De Bruijn et al., 2014; Jönsson et al., 2012; Marie et al., 2024; Seidl and Rammer, 2017; Seidl et al., 2014; Temperli et al., 2013), the capability to explicitly model historical forest damage from SBB has not yet been demonstrated at European-scale.

L160-165 negative feedback. The paragraph is not possible to understand, requires revision

As the population index has a direct negative impact on the population index in the next modelling step, we use the well-established "negative feedback" term. The paragraph has been revised for clarity after more detailed comments from referee #2 (L184).

L170 – this paragraph seems essential, but I did not manage to understand it. Suggest revising this entire section to make it understandable (and reproducible) also for a reader without deep technical understanding of this framework.

The paragraph has gone through a comprehensive revision (L194): The fraction of the different age and management classes, from the landcover functionality in LPJ-GUESS, were used to calculate  $P_{\text{gridcell}}$  weighted over the classes. With all variables in the (Marini et al., 2017) model at +/- 2 standard deviation from the mean, R has a range of -4.66 – 3.36, but interactions between variables prevent it to reach higher numbers. R calculated from the observation data used in the present study (see 2.4 below) has a range of -2.2 – 2.9, but initial high numbers in the start of an outbreak was often missing as inventories started first under an outbreak situation. The total range of R (Fig. 1a) in the presented model was set to -3.8 – 6.0, where the possible range of the different parts of the model (Eq. 4, Fig. 1a) were given weights of the similar magnitude as in the (Marini et al., 2017) model. The maximum R can also be translated to an extreme case of population increase rate with two successful generations in a year with 21 female offspring per mother ( $e^{6.1} = 21.12$ ). To enable that an outbreak can also be sustained at the highest population levels, the lowest possible total negative feedback from population size is just below the highest possible positive impact of water stress and phenology.

L198 - "that goes from zero to total shutdown of photosynthesis" - consider revising the language

This section has been substantially revised (L233): The dependency of  $P_{\text{Gridcell}}$ ,  $P_{\text{Patch}}$  and L corresponds to  $D_{\text{storm}}$  and  $D_{\text{SBB}}$  in Marini et al. (2017) model. To take advantage of LPJ-GUESS' ability to model drought impact, the Marini et al. (2017) dependency of rainfall was replaced with a dependency of the ratio between water supply to the canopy and canopy water demand (wscal), as calculated by LPJ-GUESS. The value goes from zero at complete shutdown of photosynthesis and transpiration to one at no stress, and it was used to assess the dependency of drought (1 - wscal, Fig. 1d, Eq. 4):

The authors operate across the text (already in abstract) with terms salvage logging and sanitary logging, and effect of these operations on bark beetle outbreaks. This concept can be unclear for the readers as these effects are not properly explained. Moreover, the definition of salvage (removal of windfelled trees) and sanitary (preventative removal of infested trees) is possible, but it is far from generally accepted and used definition.

We now explain the terms in the introduction (L52): To further complicate the situation, counter measures such as salvage and sanitary cutting (SSC), which are effective in preventing and shorten outbreaks of SBB, have a high variation in intensity over time and space. Salvage logging of storm

felled trees reduce the material where SBB can have very efficient regeneration and can even reduce the SBB population if the harvest is done between infection and emergence of the new generation. Sanitary cutting of infected living trees will take away the new generation, which can be a significant factor for ending an outbreak situation.

L214 infected trees. Probably infested.

We have changed to infested (L261).

L322 The sentence is not possible to understand: "To test the robustness of the approach to test the model for different parameter combinations with structure and Lmort prescribed from an LPJ-GUESS simulation with default parameters, LPJ-GUESS was finally run with the optimized parameter set, with feedback of the damage associated with that setting to the simulated vegetation". Unfortunately, such cases are frequent across the text.

We have reformulated the sentence to (L391): To test the robustness of the approach to calibrate the model for different parameter combinations with structure and  $L_{mort}$  prescribed from an LPJ-GUESS simulation with default parameters, LPJ-GUESS was finally run with the optimized parameter set, which generated the right corresponding feedback of the damage associated with that setting to the simulated vegetation.

Unclear citation in L141 (Pugh et al. Manuscript)

That manuscript is also planned to be submitted to GMD. We will update the citation with the most recent status or replace it on acceptance (L165, and also L514).

L331 "common model with the main base-run optimization" Consider please that readers main not have a deep technical understanding of this procedure

We have reformulated to (L399): Most of the parameters of the model common for all four regions in the main base-run optimization (with SSC and not including calibration data for Switzerland and Austria 2011-2019)

L333 The same as above — "It should be noted that the calibration always was based on data from all countries, also when the optimum model for the regions countries was selected, which explains why there is a difference for S Sweden and NE France between Table 3 and Table S1a and between Table S1b and Table S1c." It is necessary to find a language that makes these results accessible to and reproducible by a broader community.

Based also on comments from referee #2, we now formulate it as (L402): It should be noted that all calibrations were based on data from all countries, then the optimum model among the 77-parameter space for the regions countries, or all together, was selected, which explains why there is a difference for S Sweden and NE France when including calibration data 2011-2019 for Austria and Switzerland (Table 3 vs Table S1a and Table S1b vs Table S1c).

In this manuscript, the authors describe the implementation and performance of a module to represent spruce bark beetle infestations in the LPJ-GUESS dynamic global vegetation model (DGVM). This is important work because, while pests and disease are major drivers of forest disturbance in some regions, they are woefully underrepresented in DGVMs. I appreciate how the authors seemed to focus on building a system that is extensible for additional pest types, since the important species vary widely across the world.

The manuscript is for the most part well-written, but I do have some questions and suggestions for clarification. Similarly, the analyses are appropriate, and the figures are mostly clear. I thus recommend this to be reconsidered after what are probably minor revisions, but they are important enough that I would like to see them before they're accepted.

1. One citation of previous work is missing: Marie et al. (2024, GMD): "Simulating *lps typographus* L. outbreak dynamics and their influence on carbon balance estimates with ORCHIDEE r8627"

We have added references to the LandClim, iLand and ORCHIDEE models and rephrased the sentence referring to them to make it more specific (L57): Whilst landscape- and national-level models for SBB exist (De Bruijn et al., 2014; Jönsson et al., 2012; Marie et al., 2024; Seidl and Rammer, 2017; Seidl et al., 2014; Temperli et al., 2013), the capability to explicitly model historical forest damage from SBB has not yet been demonstrated at European-scale. And we now also explain more details of what is missing (L72): The dynamical vegetation model (DVM) ORCHIDEE has recently been updated with a mechanistic SBB functionality (Marie et al., 2024), but lacks SSC functionality. The iLand landscape simulator has mechanistic SBB components and can simulate salvage logging, but has been evaluated in protected areas to rule out the influence of SSC (Seidl and Rammer, 2017).

2. L98: Patch area is unnecessary

It has been removed (L118).

3. L125-127: Not sure what this means

We have tried to make the sentence clearer (L145): The age-class data had a regional resolution for France and a national for Austria and Switzerland. For Sweden national inventory data for 2008-2012 with a county resolution were used instead of the Poulter et al. (2019) data

4. L128: What age classes? Are those in the inventory data you're talking about? Inventory data should probably get its own introductory paragraph, before you start talking about how you incorporate forest management into LPJ-GUESS sims.

We still think it can be placed where it is but we admit that it was poorly expressed and has changed the entire sentence to (L149): To get also weights for CCF and potential natural vegetation (PNV) from the datasets, the short and long CCF classes were used to represent the fraction of the 91-110 and 111-140 years age classes in the inventory data respectively, and PNV was used to represent the fraction of forest older than 140 years.

5. L132: 5 patches per treatment seems low. Have you tested how replicable any of the results are at 5 patches, or the effect of increasing to 20 patches? It would be useful to demonstrate that 5 patches are enough for replicable results, or to increase the number of patches until it is.

Five patches may be a low number for PNV where random disturbances occur but it has a quite low total fraction. For the managed patches the random disturbances are turned off after management is initiated and there are only very minor differences, mainly in soil carbon, due to the different disturbance histories. Also, since the calibration is carried out at region/country level, the total number of gridcells involved is quite high. So, the effective number of replicates is high. We would agree with the reviewer if the calibration/evaluation was made at gridcell level, but it is not a problem at the region/country level. We have not revise this (L155).

**6. L129-141:**

a. I would use "unmanaged vegetation" rather than PNV. There are still anthropogenic effects on the unmanaged patches (e.g., population density affecting fire).

This is true, although the mentioned link of fire with population density is in fact the only such effect. However, PNV is a well-established term that has been used in many LPJ-GUESS publications in recent years. "Unmanaged vegetation" can have a temporary nature and is not crystal clear either. On this basis we prefer to keep the term PNV (L149).

b. Were patch-destroying disturbances also turned off for unmanaged patches?

No, we point the reviewer to the formulation "after the introduction of management in a patch, these were turned off" (L156).

c. Was fire turned off for any patches?

This information is now added with a new sentence (L157): Fire disturbance was simulated with the BLAZE module and it was also turned off for managed patches.

**7. L153-166:**

a. Where does the initial (t=0) bark beetle population come from?

This information has been added (L185): At start of a simulation  $P_{\text{patch}}$  was initiated with a value of 10 for all patches.

b. Is there no term to describe how population increases when a low population experiences a big surge in substrate? Or is that 2nd term on the LHS of Eq. 4 positive at low values and negative at high values? (Later, in Fig. 1, I see that the latter is the case. But you should mention this before then.)

We have made it clearer at this point (though it is later explained in conjunction with Eq. 9) by adding a comment (L188): relative to the amount of substrate with no defence (L, typically with a high value after storm damage)

**8. L168-169:**

a. What do you mean, ranges? Is this for parameterization purposes? Explain.

We now refer to equation and figure and specify by "outcome range" (L200): the possible outcome range of the different parts of the model (Eq. 4, Fig. 1a) were given weights

b. What parts of your model correspond to which parts of the Marini et al. (2017) model?

We have added (L233): The dependency of  $P_{\rm gridcell}$ ,  $P_{\rm patch}$  and L corresponds to  $D_{\rm storm}$  and  $D_{\rm SBB}$  in the Marini et al. (2017) model. To take advantage of LPJ-GUESS' ability to model drought impact, the Marini et al. (2017) dependency of rainfall was replaced with a dependency of the ratio between water supply to the canopy and canopy water demand (wscal) *The temperature part is already explained but the formulation has been revised to (L147)*: For a more mechanistic approach of taking phenology into account, the dependency of T in the (Marini et al., 2017) model was replaced with a dependency on the length of the autumn swarming period (ASP) in comparison with the grid-cell specific 30-year average as

**c. What are the ranges?**

We now try to explain and motivate our settings more clearly (L195): With all variables in the (Marini et al., 2017) model at +/- 2 standard deviation from the mean, R has a range of -4.66 – 3.36, but interactions between variables prevent it to reach higher numbers. R calculated from the observation data used in the present study (see 2.4 below) has a range of -2.2 – 2.9, but initial high numbers in the start of an outbreak were often missing as inventories only began when already under an outbreak situation. The total range of R (Fig. 1a) in the presented model was set to -3.8 – 6.0, where the possible range of the different parts of the model (Eq. 4, Fig. 1a) were given weights of the similar magnitude as in the (Marini et al., 2017) model. The maximum R can also be translated to an extreme case of population increase rate with two successful generations in a year with 21 female offspring per mother ( $e^{6.1} = 21.12$ ).

**9. L171-172: Is that realistic?**

We now acknowledge this as an extreme case (L203): The maximum R can also be translated to an extreme case of population increase rate

**10. Fig. 1a:**

a. Are the ranges used just the dark part? Or the light + dark parts?

We have revised the description for clarity (L211): For  $f(P_{gridcell})$  and  $f(P_{patch} / L)$ , the light shaded areas show the part of the ranges that were varied in the parameter optimization, where the sum of the minimum of  $f(P_{gridcell})$  and  $f(P_{patch} / L)$  were kept constant to have the possible total negative feedback from the population index constant (see section 2.6).

b. Where does the -3.8 number come from?

We now refer to section 2.6 where it is explained (L214): (see section 2.6)

**11. Fig. 1b-e:**

a. Lines are too thin, and pink is especially hard to see.

We have revised line thickness and used a darker colour of pink (L209):

b. What are the k parameters? They're not defined until after the figure. In the caption, refer the interested reader to eq. 7.

We now refer to all the equations and Table 2 and Section 2.6, where the k parameters are explained (L214): (b-e) Shape of the functions for the components of the increase rate of the bark beetle population index (R), b Eq. 7, c Eq. 8, d Eq. 10, e Eq. 12. The default parameter setting (Table 7) is shown by thick grey lines (b-d). The functions are also shown in colour for

the min and max value of parameters included in the calibration and sensitivity analysis (section 2.6), using the default setting for the other parameters.

12. L187-192: Eqs. 7 and 8 were initially confusing because I couldn't figure out why Rgridcell wouldn't just be the mean of all Rpatch values. However, these aren't actually describing the population change exponent as is implied by the use of R; they're describing different additive terms within the equation for R. (On reread, I see that the Rgridcell and Rpatch convention is introduced in Fig. 1, but that's easy to miss.) For clarity, the right-hand side of these equations should be  $f(P_{gridcell} t_{-1})$  and  $f(P_{patch t-1})/L)$ , respectively. In addition, before showing the equations, remind the reader in words what those terms are supposed to represent (respectively: effect of landscape-scale and substrate-scale competition [or the relief thereof, at low densities]).

All the equations describing the components of Eq. 4 has got their left-hand side (we assume that the reviewer has mixed left and right) replaced with the terms in Eq. 4 for consistency with that equation and Figure 1 (Eq. 7, 8, 10 and 12). We now also refer to Eq. 4 when the equations are presented.

13. It seems like the change in available material (L) is used in all these equations. I think that makes sense. But it also sounds from your text like only the POSITIVE component of L is used; e.g., L196-197. What about the NEGATIVE component—losses to fire, decomposition, and bark beetles? L should represent the NET change in substrate availability, no?

L is not the net change but the absolute amount, which is stated in the text directly after Eq. 9 (L232).

**14. L202-203:**

a. Why was wscal calculated for both the previous and current year?

As it has been shown before that the previous year can be as important. To make this clear, we have changed the formulation to (L241): The mean wscal calculated over the month May-July for both previous year (wscalt-1) and current year (wscalt) for the BNE PFT were used. Based on the (Marini et al., 2017) model, the data from the previous year were given a three times higher weight in the default setting, but in the calibration and sensitivity analysis (see section 2.4 below) the previous year weight (kpyw) was varied between  $\frac{1}{4}$  to 4:

b. This raises the question: When is this calculation happening? Is it at the end of the calendar year?

This is important information that we have missed to include, thank you. We have added this paragraph in the end of section 2.3 (L275): The bark beetle accounting and application of damage is placed in the "mortality\_guess" function in the vegdynam.cpp code together with the wind damage application, and it is called at the end of each simulated year (Lagergren et al., 2024a).

15. L205: I'm confused about how the weighted mean (wscal\_mean) is calculated. Please add an equation explaining it.

An equation has been added (L246):
$$\operatorname{wscal}_{\operatorname{mean}} = \frac{\operatorname{wscal}_{t-1} k_{\operatorname{pyw}} + \operatorname{wscal}_t}{k_{\operatorname{nyw}} + 1}$$

16. L207: How is autumn swarming period defined? A reference to Marini et al. (2017) isn't enough; for GMD you need to go into these kinds of details.

An explanation has been added (L253): where ASP is the number of flight days of the first new generation according to Jönsson et al. (2011).

**17. L214-221:**

a. When salvage and/or sanitary cutting is performed, is the prescribed harvest fraction reduced for that year? E.g., if damage\_available > salvmax, there should be no additional capacity for wood harvest in the first half of the year (when salvage/sanitary cutting is performed).

As carbon fluxes were not the focus of the present study, to simplify the simulation setup the salvage and salvage logging and sanitary cutting was not applied in the main vegetation accounting of the model and did not interfere with the prescribed harvest. We only applied it within the bark beetle outbreak calculations. We now explain this (L271): At this stage of the model development the effect of the salvage and sanitary cutting were just applied in the bark beetle accounting, the damaged trees were not removed in the main carbon accounting of the model. This can cause some underestimation of the heterotrophic respiration, but it was considered as insignificant for the present study which only focuses on SBB outbreak dynamics.

b. Where did the 5 m3 number come from?

An explanation has been added (L263): In Sweden forest owners are not allowed to leave more than 5 m3 of damaged spruce wood with d >10 cm per hectare after a storm, as regulated by the Swedish Forestry Act (Swedish Forest Agency, https://www.skogsstyrelsen.se/en/laws-and-regulations/skogsvardslagen/, last access: 12 May 2025).

**18. L252-271:**

a. It's not surprising that CRU wind data aren't very informative for wind damage, as mean wind speeds don't account for damaging gusts. For a similar finding with regard to fire Lasslop et al. (2015, DOI:10.1071/wf15052)—this would be interesting for you to note/cite.

As a response to Reviewer 1, we have instead simplified the text (L313).

b. Wouldn't it be simpler to just force with the observed storm damage? I think you don't do this because you want to account for the cohort/height-specific situation. This should be mentioned.

You are completely right, which we now explain (L315): In order to focus on bark beetle outbreak dynamics without introducing additional uncertainties associated with wind data and wind damage modelling, we prescribed wind damage from observed data. To still take advantage of the wind module's capacity to distribute wind damage among patches and cohorts' sensitivity, a calibration was done to adjust wind load (Eq. (6) in Lagergren et al., 2012) so that the observed damage was reached.

19. L276-279: Is "CRU" here referring to the CRU-JRA dataset? If so, replace all bare "CRU" references with "CRU-JRA." If not, cite the CRU dataset separately.

There was only one bare "CRU" reference and it has now been replaced with "CRU-JRA" (L344).

20. Table 2: Include a column referring to the equation(s) where each parameter is used.

A column with equation references has been added (L358).

21. L333-336: I don't understand.

We have rewritten this section (L402): It should be noted that all calibrations were based on data from all countries, then the optimum model among the 77-parameter space for the regions

countries, or all together, was selected, which explains why there is a difference for S Sweden and NE France when including calibration data 2011-2019 for Austria and Switzerland (Table 3 vs Table S1a and Table S1b vs Table S1c).

**Results**

22. L405-406: This is confusing. What is "the LPJ-GUESS run with calibrated parameters"? I thought that was what you were describing at the top of this paragraph.

This is described in section 2.7 for which we have edit the header to reflect that this test is described there: 2.7 (L290): Robustness test and exploration of the climate change signal

23. Sect. 3.4 adds nothing; it can be deleted, with any important information moved into the Discussion.

We have moved this part to the beginning of the discussion where it fits better (L490).

**Discussion**

24. L422-423: What are galleries? What is "the defense?"

We think that most readers are familiar with the gallery term but for clarity we have revised to (L501): the beetles need a dense cover of beetle larvae galleries. We have also specified to (L502): resin defence

25. L425: "tree density" initially had me thinking in terms of individuals/ha. Rephrase to "wood density" for clarity. Unless... is it actually individuals/ha? If so, please explain the connection there.

It is indeed trees per hectare, we have made this clear and explained the connection (504): The tree density (trees per ha) determines how quickly a tree reaches  $d_{lim}$ , as reduced density means increased diameter growth, and, in turn, depends on plant number, thinning and mortality.

26. L427-428: This is confusing. Each country/region had multiple gridcells, no?

As is shown in Fig. 2, there were multiple gridcells. We have now tried to make the sentence less confusing by not using the "one point in time" phrase (L507): the size of age classes was determined for one occasion at country or county scale

27. L462-465: Not sure what this bit is adding to this paragraph.

We will keep it but have revised it to make it clearer what is challenging for dynamic forest models (L544): Many SBB models require an initial population or damage level, which means that they work in relative terms (Marini et al., 2017; Soukhovolsky et al., 2022), but a dynamic forest model needs to operate with absolute damage levels making the modelling more challenging.

28. L474: "but" doesn't seem to fit here.

We have revised to (L556): but in recent year it has also triggered as well as contributed to sustained outbreaks of SBB in Europe (Nardi et al., 2023; Trubin et al., 2022).

29. L493-495: How do # of generations emerge from the phenology function, which seems to just be Eq. 14?

We now motivate the use of ASP instead of number of generations in connection to Eq. 12 (L256): The dependency of ASP instead of, e.g., the number of generations per year was chosen as ASP is a continuous variable which better catch the average when there is a variability in the climate, such in mountainous regions, then a discrete variable. We have also acknowledged and reminded the readers that we use ASP and not number of generations (L578): The SBB phenology response, which is a function of the length of the first generation's swarming period, also reaches a plateau when the climate is warm enough to allow a complete second generation to emerge every year.

---

## Referee Report (RR1)

Review of revision 1: "Combining empirical and mechanistic understanding of spruce bark beetle outbreak dynamics in the LPJ-GUESS (v4.1, r13130) vegetation model" (gmd-2024-239)

I was reviewer 2 on the initial manuscript; the authors have done a good job responding to my initial comments. I have some more, but I recommend this be published subject to what I think are minor revisions. The only reason I think I would need to review again would be if there still seems to be confusion about my comment 13 on my original review (which I'll refer to as my comment 1.13; see comment 2.6 below).

All line numbers refer to the "authors' tracked changes" version of the manuscript.

**Comments on replies to my initial review (mostly):**

- 2.1) Re: my comment 1.6: I think I'm satisfied with disturbance being turned off in this experiment, given the likely low rate in this region.
- 2.2) L194-201: "The fraction of the different age and management classes, from the landcover functionality in LPJ-GUESS, were used to calculate Pgridcell weighted over the classes. With all variables in the (Marini et al., 2017) model at +/2 standard deviation from the mean, R has a range of -4.66 3.36, but interactions between variables prevent it to reach higher numbers. R calculated from the observation data used in the present study (see 2.4 below) has a range of -2.2 2.9, but initial high numbers in the start of an outbreak were often missing as inventories only began when already under an outbreak situation. The total range of R (Fig. 1a) in the presented model was set to -3.8 6.0, where the possible outcome range of the different parts of the model (Eq. 4, Fig. 1a) were given weights of the similar magnitude as in the (Marini et al., 2017) model."
  - o Refer to Sect. 2.6 where you calculate the -3.8 number.
  - What are the "weights" here that you are "giv[ing]" the model components? I
    don't see any weights in Eq. 4, and the only weights previously mentioned
    have to do with the different age/management classes.
  - Is the "The total range of R" another setting you applied, or is it just the emergent result of the "weights" you gave?
- 2.3) Fig. 1a: Are the light parts the *only* ranges you used for optimization, or did you also include the dark part? And what is the dark part—the final allowed range?
- 2.4) Fig. 1b: Was k2 min intentionally changed from pink to orange? If so, why?
- 2.5) Before showing Equations 7 and 8, remind the reader in words what those terms are supposed to represent (respectively: effect of landscape-scale and substrate-scale competition [or the relief thereof, at low densities]).

- 2.6) Re: my comment 1.13: I don't think I was clear. Looking at Eq. 9, and the text directly after (as the authors mentioned in their reply), I'm only seeing change in L due to tree mortality: "Lmort is C mass of stem mortality of spruce trees above a diameter threshold (dlim) for previous year caused by other reasons than bark beetles (including storm)." That's not an "absolute amount" as the authors said in their reply; the way I read that sentence, Lmort is one influx to L. That's because mortality is a flux variable, not a state variable. Why would L only include one year's influx rather than the total amount of stem litter? Do beetles not eat wood that's been dead for more than a year?
- 2.7) Re: my comment 1.24: The authors reply that they "think that most readers are familiar with the gallery term," but least 50% of the readers so far (i.e., me) weren't. I'm a vegetation modeler with no entomological background, which I suspect describes many of this article's potential readers. It should be defined.
- 2.8) Re: my comment 1.27: I guess what I was saying is that I don't understand how this bit relates to the rest of the paragraph. It would be good for the authors to draw that connection.
- 2.9) Re: my comment 1.28: Still, "but" seems wrong. The part of the sentence after the comma seems to be *supporting* the part of the sentence before, rather than contradicting it.

**Other comments**

- 2.10) L75: "but has **only** been evaluated"
- 2.11) L87: "are" should be "is".
- 2.12) L157: "BLAZE" should be "SIMFIRE-BLAZE": BLAZE is just the fire impacts module, with SIMFIRE giving burned area. There should be a citation here of the first LPJ-GUESS SIMFIRE paper: Knorr et al. (2016, *Nat. Clim. Chg.*, doi: 10.1038/nclimate2999). BLAZE unfortunately hasn't been published; the most complete description is as part of Rabin et al. (2017, *GMD*, doi: 10.5194/gmd-10-1175-2017).
- 2.13) L186: "R for M (Eq. 5)" should be "M on R (Eq. 6)".
- 2.14) L362: "where" should be "were".
- 2.15) L483: "spruce" should be "spruce's".
- 2.16) L490: Delete "of a".
- 2.17) L567-569: This new sentence is too vague, or maybe it's hard to understand why it's included. It needs to be tied in more clearly with the story being told in this paragraph.
- 2.18) L575: "has" should be "have".

---

## Author Response (AR2)

Our comments are written *in italic red font*. Revised text is written in plain red text, for which we give reference to the beginning line number (*LXXX*) of the revised annotated version. Also, note that we have used EndNote for handling references and that changes to those do not always show up as revised.

**Topic editor decision: Reconsider after major revisions**

by Roslyn Henry

**Public justification** (visible to the public if the article is accepted and published):

The revisions have improved the paper however further revisions are required to adequately address previous reviewer comments with regards to clarity.

**Additional private note (visible to authors and reviewers only):**

The revisions required are not substantial but important nonetheless.

Improving the clarity of the paper following reviewer comments and addressing reviewer #1 comment regarding the discussion ("My previous criticism of the discussion, which contained general information about bark beetles rather than proper discussion of results, was not addressed. The overall revision of Discussion was minor/formal though it required rather thorough update.") is essential with this round of revisions.

Both myself and the reviewers find great value in this work and a more careful addressing of comments will see this work publishable.

Thanks for giving us the opportunity to revise and improve the manuscript, we have tried to put a good effort in properly discussing our results and generally clarifying the text.

**Anonymous referee #1 report**

- 1) Scientific significance: Does the manuscript represent a substantial contribution to modelling science within the scope of this journal (substantial new concepts, ideas, or methods)? **Good**
- 2) Scientific quality: Are the scientific approach and applied methods valid? Are the results discussed in an appropriate and balanced way (consideration of related work, including appropriate references)? Do the models, technical advances and/or experiments described have the potential to perform calculations leading to significant scientific results? Good
- **3)** Scientific reproducibility: To what extent is the modelling science reproducible? Is the description sufficiently complete and precise to allow reproduction of the science by fellow scientists (traceability of results)? **Fair**
- **4) Presentation quality:** Are the scientific results and conclusions presented in a clear, concise, and well structured way (number and quality of figures/tables, appropriate use of English language)? **Poor**

For final publication, the manuscript should be - reconsidered after major revisions

Were a revised manuscript to be sent for another round of reviews: I would be willing to review the revised manuscript.

**Suggestions for revision or reasons for rejection**

(visible to the public if the article is accepted and published)

Dear Editor, Dear Authors,

I read the revised paper several times and I am afraid I still cannot recommend it for publication. In my opinion, revisions made in response to my first review were not adequate. Parts of the manuscript remain overly technical, sometimes written in a tone of developer's guidelines rather than scientific text. Conceptually, the logic of defining different process complexity across the model still remain unclear (see also me previous comments), without justifying this approach. I do not feel that the current response and the revision addressed this comment (L110, .... adopting empirical elements wherever possible). We have completely rewritten this last part of the introduction to make it clearer what we aim for (L117): To achieve this, a semi-empirical SBB damage model, with components of negative feedback from a dense SBB population, amplification of damage after storm felling, where warm and dry weather can trigger and contribute to sustained outbreaks and SSC functionality was developed. The model was calibrated based on storm and SBB damage statistics from four countries in Europe. This modelling concept represents key aspects of the interaction of SBB with climate and forest state, creating a tool for addressing climate change and forest management scenarios.

Most of my comments from the previous review round are still valid (I will justify this through a few cases below). Most of the revisions are formal (adding a sentence) rather than trying to solve the commented problem. Sorry for this. We have now tried to take a more thorough evaluation of all comments.

My previous criticism of the discussion, which contained general information about bark beetles rather than proper discussion of results, was not addressed. The overall revision of Discussion was minor/formal though it required rather thorough update. We have gone through the discussion, rewritten sections of it, removed some more general material and made sure that all sections have references to our results. We noted that especially the beginning of the discussion tended towards being overly general and have especially focused our revisions there. We believe the later paragraphs are generally already closely tied to the results of our modelling.

Here I indicate just a few cases:

- Revision made in L55 "... To further complicate the situation ...." uses inappropriate terminology (regeneration should probably be reproduction, infection -> infestation, etc.), which I already comment on in the first review. We have gone through the entire manuscript and corrected the terms (L61, L62, L494, L617). It also lacks any references to literature. Generally, scientific evidence that SSC can shorten and prevent the outbreak are rather unconvincing. We have added several references regarding the impact of SSC throughout the manuscript, and have also tried to give a wider view of the uncertainty of the effect in the introduction (L58): To further complicate the situation, counter measures such as salvage and sanitary cutting (SSC), which can be effective in preventing and shortening outbreaks of SBB (Jönsson et al., 2012; Stadelmann et al., 2013), have a high variation in

intensity over time and space. Salvage logging of storm felled trees reduce the material where SBB can have very efficient reproduction and can even reduce the SBB population if the harvest is done between infestation and emergence of the new generation. Sanitary cutting of infested living trees will take away the new SBB generation if it is done at the first signs of infestation in summer. This can contribute to ending an outbreak situation. Sanitary cutting is less efficient when done in autumn or winter as only a fraction of the new beetles is still present (Singh et al., 2024). In M&M (L302): As the numbers for the SSC setting vary over time, regions and countries (Jönsson et al., 2012; Stadelmann et al., 2013; Wichmann and Ravn, 2001), we do not claim that we have found the most representative numbers for all of Europe but they rather should be seen as a starting point for evaluating SSC dependency in this modelling concept. And in the discussion (L616): In managed forest, however, the occurrence and timing of counter measures such as salvage logging of storm-felled trees, sanitary cutting of infested trees and insect traps can have a big impact on the outcome (Jönsson et al., 2012), factors that vary greatly in time and by region over Europe, as well as at finer scale (Wichmann and Ravn, 2001). Stadelmann et al. (2013), e.g., reports that higher fractions of wind damage were associated with lower intensity of salvage logging after 1999 storm Lothar in Switzerland, while after the Gudrun 2005 storm in Sweden extra resources were brought in from other parts of the country, as well as from abroad, to increase the rate of salvage cutting (Fridh, 2006). By timely salvage logging, a large fraction of the beetles can be removed from the forest, substantially dampening the damage done (Jönsson et al., 2012; Stadelmann et al., 2013). An indirect indicator of the effect of counter measures is that the risk for SBB infestations are higher with nature reserves in the landscape where no SSC is done (Kärvemo et al., 2023).

- Revision made in L110 justifying the philosophy of development. As I wrote above, this does not really helped to clarify my comment from my previous review. In all honesty we are struggling with this one. Throughout paragraphs 2 (L42) and 3 (L58) of the introduction (revised version – previously one paragraph) we explain the important characteristics governing bark beetle outbreaks. In Paragraph 4 we outline previous process-based and empirical efforts to model bark beetle impacts and their strengths and weaknesses. Then in Paragraph 5 (L97) we outline the characteristics we are looking for in a model, including a list of what it should do. Clearly a model with uniform processcomplexity would appear elegant but would not take account of simple realities; there are some things we know relatively well and can constrain at relatively high complexity – e.g. beetle phenology - and some things we consider conceptually very important to be able to explore scenarios, whilst other things we know less well and consider less conceptually important. We have made a change to the start of Paragraph 5 to put more upfront what the SBB model is intended to be used for and matched this with a new sentence at the end of the paragraph. We have also added references in the model description to why we have gone for a particular level of complexity for different components as we introduce them (see e.g. Paragraph 2 of Section 2.3; L193): This additive model concept captures the dynamics of SBB outbreaks based on a tractable set of process-linked predictors that are empirically-supported at large scales. We therefore found this to be a strong basis for our approach and implemented an additive model drawing on this concept for bark beetle damage in LPJ-GUESS. In this implementation we took advantage of the existing LPJ-GUESS formulations for water stress and mortality of trees (L, stem litter from spruces larger than a limit ( $d_{lim}$ ) killed by other agents than bark beetles last year) as inputs, as well as the available and proven bark-beetle phenology sub-model of Jönsson et al. (2007). We hope that these steps make the reasoning for the choices clearer.

**- My previous comments:**

(i) "Does this approach have any justification? To enable that an outbreak can also be sustained at the highest population levels ...;

(ii) "I have a similar concern as in the previous point concerning the implementation of salvage logging ......".

I do not consider the way these comments were addressed adequate. Concerning the former comment, there was just a single word change in this sentence (feedback to impact, L205). We have revised the sentence to make it more general and added a reference (L228): To enable that an outbreak can be sustained also during the epidemic phase with the highest population levels (Hlásny et al., 2021), the lowest possible total negative feedback from population size was set to be just below the highest possible positive impact of water stress and phenology. In response to referee #2 we have also made clarifications to the entire section and Figure 1 (see specific referee#2 comment 2.2 below). Concerning the latter comment, it concerned the lack of justification on real dampening effect of salvage logging on outbreak in the presented implementation. In the revision, the comments was addressed just by referring to the Swedish Forestry Act, which cannot be considered as a relevant scientific source. As written above, we have now added several references regarding the impact of SSC throughout the manuscript and have also tried to give a wider view of the uncertainty of the effect (see comment above with reference to L58, L302 and L616).

- My previous comment: .. It is really extremely difficult to understand the logic of such statement: "As storms in Europe mainly occur in autumn and winter ... I did not find the revision helpful, the sentence still remains extremely complicated. We have divided the sentence to make the message clearer. We also changed from "storm season" to "storm year" and are referring to the well-established "water year" term that has a similar usage (L324): Storms in Europe occur mainly in autumn and winter, and the modelled vegetation and bark beetle effect, e.g., should be the same for a storm event in October or in February the next year. The storm damage statistics for a specific year were, therefore compiled for a storm year (c.f. the commonly used "water year" term, Johnstone and Cross, 1949) of 12 months from July until June the next year when building the dataset used for the calibration.
- My previous comment "As far as I understand this part, the authors fixed a poor match of simulated wind damage with observations by introducing a correction by latitude, which correlated with productivity in Sweden (unpublished data?), and because there is higher damage in more productive sites, it should help simulate wind damage better. If this is correct, it looks like rather artificial solution for improving model performance. If the wind module is coupled with the vegetation model, should not the productivity and subsequently wind susceptibility be simulated as an emergent property? Without a need to imprint there this pattern externally" was not addressed by the current response. The point was why productivity gradient needs to be imprinted into the simulations through latitudinal gradient, if vegetation dynamics model is part of the implementation. The current response just says that productivity is highly correlated with latitude, based on the Swedish forest statistics. We recognise that the discussion of latitude as a proxy of productivity gave the impression that we intended this calibration as something more than simply adjusting wind damage to the observed level in order to focus on the bark beetle calibration. We have thus removed that text and added a sentence to clarify this (L355): As we wished to use the existing wind module's capability to distribute wind damage among cohorts' in different patches depending on their sensitivity (Eq. 1), the model could not be driven directly by the available observations of DFstorm. Therefore, instead of calculating WL from the cubed exceedance of the 99.5 percentile of daily wind speed accumulated over storm season as in Lagergren et al. (2012), a calibration was done to adjust WL so that modelled damage followed the observed DFstorm (denoted WLstat). We have also made some additional revision to the section for clarity (L373): As a common linear scaling was used to go from DFstorm to WLstat, the exact DFstorm time series will not be reproduced by this approach, but the pattern and level should be

reasonable well captured. The WLstat time series were then used as external input to the model runs. This procedure was followed purely to provide an observationally consistent wind damage level for the calibration of the SBB model and is not intended as a European-scale parameterisation of the Lagergren et al. (2012) wind module, which will be carried out in a separate study.

Moreover, language revision is highly needed, many sentences does not read well or are inaccurate. We have checked the language for the entire manuscript. In abstract, for example:

- L10 Tree mortality is an important function in such models and, especially for needle leaved forest in the temperate and boreal zones, bark beetles are important for the mortality pattern. (quite difficult to read). The first three sentences have been revised (L9): For exploring forest performance in the future, dynamic vegetation models are important tools. Tree mortality is a crucial process in these models, but explicit representations of major agents of mortality have often been relatively underdeveloped. In needle-leaved forest in the temperate and boreal zones, bark beetles are often important for the mortality pattern.
- L15 Drought can contribute to increased damage and prolonged outbreaks by lowering the defence of the trees, but has been the main driver of some of the European forest damage in the last decade. (I cannot understand what this sentence communicates). We have made clarifications to the sentence (L15): Drought can contribute to increased damage and prolonged outbreaks by lowering the defence of the trees, but it has also been the main ultimate driver of some of the European forest damage by bark beetles in the last decade.
- L20 An index of the SBB population size that changed over time driven by phenology, drought, storm felled spruce trees and density of the beetle population, was used to scale modelled damage. (I also needed to read the sentence repeatedly. Why not to write directly ... The modelled damage was driven by bark beetle phenology ... ). We have revised accordingly (L22): The model was driven by SBB phenology
- ... but there were discrepancies in levels, which partly can be related to salvage logging of storm felled forest and sanitary cutting of infested trees. (I did not find in the paper any proof that the discrepancy was caused by salvage and sanitation logging; the reason can also be limitations in the implementation or in general understanding of processes, etc.). In Figure 5 we show that the model results are very sensitive to the setting of SSC. And with the subtler presentation of the SSC uncertainty (see comments above), we have only made a minor modification to this sentence (L23): The model was able to catch the start and duration of outbreaks triggered by storm damage reasonably well, but with discrepancies in levels which can be at least partly related to salvage logging of storm felled forest and sanitary cutting of infested trees.
- .... which may suggest that the drought stress response of spruce in LPJ-GUESS is underestimated (models generally suffer from limited connection of tree stress physiology and biotic agents, which is more complex that just stress underestimation, see e.g., <a href="https://doi.org/10.1016/j.tree.2021.02.001">https://doi.org/10.1016/j.tree.2021.02.001</a>). We removed the word "stress" here in the abstract (L27) and now mention this complexity, using the suggested reference, in the discussion (L659): It is also clear that physiological drought modelling is a challenge (Trugman et al., 2021).
- L50 not true, thinning, on the contrary, reduces stand density and increases resistance to beetles. *Yes that is right, but the thinning effect is much more complicated than that, which we now explain (L53):* Thinning may increase the trees' resistance by reducing competition (Singh et al., 2024) but, on the other hand, it can cause direct damage by the logging operation (Hwang et al., 2018), introduce root rot (Vollbrecht and Agestam, 1995) and make the stand more prone to wind damage (Nielsen,

1995). It also increases the growth rate of the remaining trees, making them accessible to SBB at a younger age.

Also later in the text, many sentences are very difficult to read, are overly long and complex, often using technical jargon. E.g.:

- L257: The dependency of ASP instead of, e.g., the number of generations per year was chosen as ASP is a continuous variable which better catch the average when there is a variability in the climate, such in mountainous regions, then a discrete variable. (impossible to understand). *The sentence has been revised (L284):* The use of ASP instead of a discrete variable such as the number of generations per year, was chosen as ASP is a continuous variable which better catch the average conditions within a climate grid cell when there is a high variability in temperature, such as in mountainous regions. Furthermore, a well-defined model with constrained parameters already existed for ASP (Jönsson et al., 2011).

In my opinion, the manuscript contains a substantial amount of good work and valuable results. However, the reproducibility and clarity of this work for a broader readership remains limited. Therefore, I cannot recommend its publication in the current form.

**Anonymous referee #2 report**

- 1) Scientific significance: Does the manuscript represent a substantial contribution to modelling science within the scope of this journal (substantial new concepts, ideas, or methods)? **Good**
- 2) Scientific quality: Are the scientific approach and applied methods valid? Are the results discussed in an appropriate and balanced way (consideration of related work, including appropriate references)? Do the models, technical advances and/or experiments described have the potential to perform calculations leading to significant scientific results? Excellent
- **3)** Scientific reproducibility: To what extent is the modelling science reproducible? Is the description sufficiently complete and precise to allow reproduction of the science by fellow scientists (traceability of results)? **Excellent**
- **4) Presentation quality:** Are the scientific results and conclusions presented in a clear, concise, and well structured way (number and quality of figures/tables, appropriate use of English language)? **Good**

For final publication, the manuscript should be - accepted subject to minor revisions

Were a revised manuscript to be sent for another round of reviews: I would be willing to review the revised manuscript.

**Suggestions for revision or reasons for rejection**

(visible to the public if the article is accepted and published)

I was reviewer 2 on the initial manuscript; the authors have done a good job responding to my initial comments. I have some more, but I recommend this be published subject to what I think are minor revisions. The only reason I think I would need to review again would be if there still seems to be confusion about my comment 13 on my original review (which I'll refer to as my comment 1.13; see comment 2.6 below).

All line numbers refer to the "authors' tracked changes" version of the manuscript.

**Comments on replies to my initial review (mostly):**

- 2.1) Re: my comment 1.6: I think I'm satisfied with disturbance being turned o in this experiment, given the likely low rate in this region. *Good*.
- 2.2) L194-201: "The fraction of the different age and management classes, from the landcover functionality in LPJ-GUESS, were used to calculate Pgridcell weighted over the classes. With all variables in the (Marini et al., 2017) model at  $\pm$ 2 standard deviation from the mean, R has a range of  $\pm$ 4.66 3.36, but interactions between variables prevent it to reach higher numbers. R calculated from the observation data used in the present study (see 2.4 below) has a range of  $\pm$ 2.2 2.9, but initial high numbers in the start of an outbreak were often missing as inventories only began when already under an outbreak situation. The total range of R (Fig. 1a) in the presented model was set to

- 3.8 6.0, where the possible outcome range of the different parts of the model (Eq. 4, Fig. 1a) were given weights of the similar magnitude as in the (Marini et al., 2017) model."
  - Refer to Sect. 2.6 where you calculate the -3.8 number. The -3.8 number is actually not calculated in 2.6 but is motivated and based on the reasoning on L215-224, which we have tried to clarify further: In the Marini et al. (2017) model with all variables set at +/- 2 standard deviation from the mean, R has a range of -4.66 – 3.36. Interactions between variables prevent R from reaching higher numbers. R calculated in this way from the observation data used in the present study (see 2.4 below) has a range of -2.2 – 2.9. It should be considered, however, that in both cases initial high numbers in the start of an outbreak were often missing as inventories only began when already under an outbreak situation. In the present application, R is applied at patch level while observed data have been aggregated over large regions or countries, a minimum number closer to the Marini et al. (2017) value is therefore motivated. At the other end of the scale, the maximum R can also be translated to an extreme case of population increase rate with two successful generations in a year with 20 female offspring per mother ( $e^6 = 20.12$ ). We have also clarified the base for this setting in the next sentence (L224): Based on this, the total possible range of R (Fig. 1a) in the presented model was set to -3.8 - 6.0, where the possible outcome range of the different parts of the model (Eq. 4, Fig. 1a) were given weights (relative contribution to the total range) of the similar magnitude as in the Marini et al. (2017) model.
  - What are the "weights" here that you are "giv[ing]" the model components? I don't see any
    weights in Eq. 4, and the only weights previously mentioned have to do with the different
    age/management classes. We now explain what we mean by "weights" in this context (L226):
    (relative contribution to the total range)
  - Is the "The total range of R" another setting you applied, or is it just the emergent result of the "weights" you gave? We have made some clarifications to this sentence (L224): Based on this, the total possible range of R (Fig. 1a) in the presented model was set to -3.8 - 6.0, where the possible outcome range of the different parts of the model (Eq. 4, Fig. 1a) were given weights (relative contribution to the total range) of the similar magnitude as in the Marini et al. (2017) model. And also to the Figure 1 text (L235): Figure 1: The different components of the increase rate of the bark beetle population index (R). Depending on the state of the model and climate, R and its components take a value within the possible ranges. (a) Possible ranges of the components and the total (Eq. 4). For  $f(P_{gridcell})$  and  $f(P_{patch} / L)$ , the light shaded areas show the part of the ranges that were varied in the parameter optimization, where the sum of the minimum of  $f(P_{gridcell})$  and  $f(P_{patch} / L)$  was kept constant to have the possible total negative feedback from the population index constant (illustrated with the dotted lines, see section 2.6). (b-e) Shape of the functions for the components of the increase rate of the bark beetle population index (R), b Eq. 7, c Eq. 8, d Eq. 10, e Eq. 12. The default parameter setting (Table 7) is shown by thick grey lines (b-d). The functions are also shown in colour for the min and max value of parameters included in the calibration and sensitivity analysis (section 2.6), using the default setting for the other parameters. For f(phenology), (e), no parameters were tested but the response depends on the grid-cell's 30-year running mean of the length of the autumn swarming period (ASP, ASP30) and function are shown for ASP30 from 2 to 75 days. We also added the word "possible" to the upper bar in figure 1a.
- 2.3) Fig. 1a: Are the light parts the only ranges you used for optimization, or did you also include the dark part? And what is the dark part—the final allowed range? If  $k_{gc\_min}$  is at its max value then  $k_{p\_min}$  is at its min value and vice versa, we have tried to make this clearer by connecting those points by dotted lines in Figure 1a, and it is further explained in the figure text (**L238**): the sum of the

minimum of  $f(P_{\rm gridcell})$  and  $f(P_{\rm patch} / L)$  was kept constant to have the possible total negative feedback from the population index constant (illustrated with the dotted lines, see section 2.6) and is later further explained (L408): To further reduce the number of calculations and to keep the total weight of the population size dependency constant, the sum of the minimum of the ranges  $f(P_{\rm gridcell})$  and  $f(P_{\rm patch} / L)$  was kept constant ( $k_{\rm gc\_min} + k_{\rm p\_min} = -3.8$ ).

- 2.4) Fig. 1b: Was k2 min intentionally changed from pink to orange? If so, why? Thanks for being observant, this was a mistake,  $k_2$  min is now pink.
- 2.5) Before showing Equations 7 and 8, remind the reader in words what those terms are supposed to represent (respectively: effect of landscape-scale and substrate-scale competition [or the relief thereof, at low densities]). We now do this (L247): The representation of landscape scale and substrate scale competition (or relief of competition at low densities) was formulated in two equations (Eq. 7 and 8), respectively.
- Re: my comment 1.13: I don't think I was clear. Looking at Eq. 9, and the text directly 2.6) after (as the authors mentioned in their reply), I'm only seeing change in L due to tree mortality: "Lmort is C mass of stem mortality of spruce trees above a diameter threshold (dlim) for previous year caused by other reasons than bark beetles (including storm)." That's not an "absolute amount" as the authors said in their reply; the way I read that sentence, Lmort is one influx to L. That's because mortality is a flux variable, not a state variable. Why would L only include one year's influx rather than the total amount of stem litter? Do beetles not eat wood that's been dead for more than a year? From this comment it is clear that we need to explain how the bark beetles live in the bark. The beetles do not eat wood and they can not breed in trees that has been dead for more than 1-2 years, which we now clarify. By this it was also natural to define what a "gallery" is (comment 2.7 below) (L259): The beetle larvae feed from the phloem in galleries under the bark (Six and Wingfield, 2011). The amount of phloem depends on thickness and area of the bark, and is closely related to stem biomass. The trees are normally no longer suitable for breeding one or two seasons after tree death (Göthlin et al., 2000; Louis et al., 2014), which is why  $L_{mort}$  is based only on the mortality of the previous year.
- 2.7) Re: my comment 1.24: The authors reply that they "think that most readers are familiar with the gallery term," but least 50% of the readers so far (i.e., me) weren't. I'm a vegetation modeler with no entomological background, which I suspect describes many of this article's potential readers. It should be defined. We now explain this in M&M (see comment 2.6 above).
- 2.8) Re: my comment 1.27: I guess what I was saying is that I don't understand how this bit relates to the rest of the paragraph. It would be good for the authors to draw that connection. We have now tried to make a better flow of the last part of the paragraph connecting the sentences (1641): The outcome after a trigger also depends on the initial level of the SBB population, but many SBB models require an initial population or damage level, which means that they only need to work in relative terms (Marini et al., 2017; Soukhovolsky et al., 2022). A dynamic forest model, on the other hand, needs to operate with absolute damage levels making the modelling more challenging.
- 2.9) Re: my comment 1.28: Still, "but" seems wrong. The part of the sentence after the comma seems to be supporting the part of the sentence before, rather than contradicting it. We get the point and have moved some parts after the comma to beginning (L659): Warm weather accompanied by drought is often seen as a factor contributing to sustained outbreaks (e.g. Bakke, 1983), but in recent year it has also triggered outbreaks of SBB in Europe (Nardi et al., 2023; Trubin et al., 2022).

**Other comments**

- 2.10) L75: "but has **only** been evaluated". Fixed (L84).
- 2.11) L87: "are" should be "is". Fixed (L100).
- 2.12) L157: "BLAZE" should be "SIMFIRE-BLAZE": BLAZE is just the fire impacts module, with SIMFIRE giving burned area. There should be a citation here of the first LPJ-GUESS SIMFIRE paper: Knorr et al. (2016, Nat. Clim. Chg., doi: 10.1038/nclimate2999). BLAZE unfortunately hasn't been published; the most complete description is as part of Rabin et al. (2017, GMD, doi: 10.5194/gmd-10 1175-2017). *References added (L172):* Fire disturbance was simulated with the SIMFIRE-BLAZE module (Knorr et al., 2016; Rabin et al., 2017), and it was also turned off for managed patches.
- 2.13) L186: "R for M (Eq. 5)" should be "M on R (Eq. 6)". What is actually right is "M on Ppatch (Eq. 6)", which we now write (**L205**):
- 2.14) L362: "where" should be "were". Fixed (L405).
- 2.15) L483: "spruce" should be "spruce's". Fixed (L528).
- 2.16) L490: Delete "of a". Fixed (L532).
- 2.17) L567-569: This new sentence is too vague, or maybe it's hard to understand why it's included. It needs to be tied in more clearly with the story being told in this paragraph. We have edited the section to make it clearer how the reference fits in the context (L670): This may be a failing of the model parameterization for water stress or of the input climate forcing dataset (Steinkamp and Hickler, 2015). Similar results were also found for the most recent application of SBB damage with the ORCHIDEE vegetation model, concluding a shortcoming linked to high damage levels associated with extreme drought (Marie et al., 2024).
- 2.18) L575: "has" should be "have". Fixed (L680).

---

## Author Response (AR3)

**Remarks from the preceding review file validation**

With the next revision, please add the DOIs https://doi.org/10.5281/zenodo.14411974 and https://doi.org/10.5281/zenodo.14415079 to the "in-text" citations (Lagergren et al., 2024a) and (Lagergren et al., 2024b) in the sections "Code availability" and "Data availability". We have added the DOI links for the LPJ-GUESS code (Line 578-579) and the model results (L582-583), and we have also got a DOI for the deposit of the scripts used for the analysis and figure production (L579-580).